# Revealing broken valley symmetry of quantum emitters in WSe$_2$ with chiral nanocavities

Longlong Yang [1,2], Yu Yuan[1,2], Bowen Fu[3], Jingnan Yang[3], Danjie Dai[1,2], Shushu Shi[1,2], Sai Yan[1,2], Rui Zhu[1,2], Xu Han[4], Hancong Li[3], Zhanchun Zuo [1,2], Can Wang [1,2,5] ✉, Yuan Huang [4] ✉, Kuijuan Jin [1,2,5], Qihuang Gong [3,6] & Xiulai Xu [3,6] ✉

Single photon emission of quantum emitters (QEs) carrying internal degrees of freedom such as spin and angular momentum plays an important role in quantum optics. Recently, QEs in two-dimensional semiconductors have attracted great interest as promising quantum light sources. However, whether those QEs are characterized by the same valley physics as delocalized valley excitons is still under debate. Moreover, the potential applications of such QEs still need to be explored. Here we show experimental evidence of valley symmetry breaking for neutral QEs in WSe$_2$ monolayer by interacting with chiral plasmonic nanocavities. The anomalous magneto-optical behaviour of the coupled QEs suggests that the polarization state of emitted photon is modulated by the chiral nanocavity instead of the valley-dependent optical selection rules. Calculations of cavity quantum electrodynamics further show the absence of intrinsic valley polarization. The cavity-dependent circularly polarized single-photon output also offers a strategy for future applications in chiral quantum optics.

Single-photon emitters in the atomically thin transition-metal dichalcogenides layers have caught numerous attention in recent years as promising platforms for future high-performance quantum light sources[1–6], indicating the potential applications in quantum information processing[7,8]. Owning to its unique two-dimensional property, such emitters can be directly engineered by the local strain field[9–11] or integrated with various optical cavities and waveguides[12–17]. Normally, quantum emitters (QEs) in 2D monolayers are believed to be excitons spatially localized by defects. One question in the study of QEs is whether the excitons trapped in the defect potential inherit the valley properties of the delocalized two-dimensional exciton in band gap,

such as spin-valley locking and valley polarization[18–22], which lays the foundation for quantum emitters being used in valley-based information processing[1]. If QEs inherit these properties, they can be considered as valley excitons, enabling a range of valley-related applications[18,23]. However, if these properties are not inherited, it is crucial to investigate the precise origin and intrinsic physical properties of QEs. This will facilitate their applications in quantum light sources and integrated photonics with flexible external control because of their exceptional optical properties and two-dimensional characteristics.

It has been argued that both the anisotropic responses of QEs in layer to a magnetic field applied in-plane versus out-of-plane and the

[1]Beijing National Laboratory for Condensed Matter Physics, Institute of Physics, Chinese Academy of Sciences, Beijing 100190, China. [2]CAS Center for Excellence in Topological Quantum Computation and School of Physical Sciences, University of Chinese Academy of Sciences, Beijing 100049, China. [3]State Key Laboratory for Mesoscopic Physics and Frontiers Science Center for Nano-optoelectronics, School of Physics, Peking University, 100871 Beijing, China. [4]Advanced Research Institute of Multidisciplinary Science, Beijing Institute of Technology, Beijing 100081, China. [5]Songshan Lake Materials Laboratory, Dongguan, Guangdong 523808, China. [6]Peking University Yangtze Delta Institute of Optoelectronics, Nantong, Jiangsu 226010, China. ✉e-mail: canwang@iphy.ac.cn; yhuang@bit.edu.cn; xlxu@pku.edu.cn

cross-circularly polarized emissions of Zeeman splitting peaks at a large magnetic field (as shown in Fig. 1a) give evidence that the QEs inherit the valley physics and share the band structure of two-dimensional excitons[24,25]. Experiments on the broadband with multiple defects in WSe₂ monolayer[26] and calculations based on intravalley defects[27] have shown a polarization similar to that of valley excitons. Although positively charged QEs in WSe₂ have been demonstrated experimentally with a valley polarization, but that has not been observed in neutral excitons[24]. In addition, the *g*-factors of QEs under an out-of-plane magnetic field (-8–13)[1,3,28] are several times larger than that of valley excitons (~4.4)[21] in WSe₂ layer. Therefore, the precise origin of excitons for QEs is still not clear.

Recently, a microscopic model for emitters in WSe₂ has been theoretically proposed, which is based on the recombination of intervalley defect excitons arising from the hybridization of the dark excitons with intragap defect states[29,30]. The model has well explained

the strong brightness and hyperfine magneto-optic properties of these emitters, as shown in Fig. 1b. In the presence of intragap defect level and external strain, the dark excitonic conduction band hybridizes with the localized defect states, forming a new intervalley defect exciton state with transitions from the defect levels to two valence bands. In contrast to valley excitons, the hybridization with the defect state not only breaks the spin-valley locking, but also dramatically increases the transition oscillation strength, resulting in efficient photon emission.

The intervalley excitonic defect states in the layer suggest that the optical circular dichroism for the QEs transitions is no longer determined by the valley-dependent optical selection rules, that is, the intrinsic valley protection is absent. However, this issue has not been experimentally confirmed until now, especially in the case of a single quantum emitter. Furthermore, the potential applications of the intervalley defect excitons remain to be explored. Optical cavity can

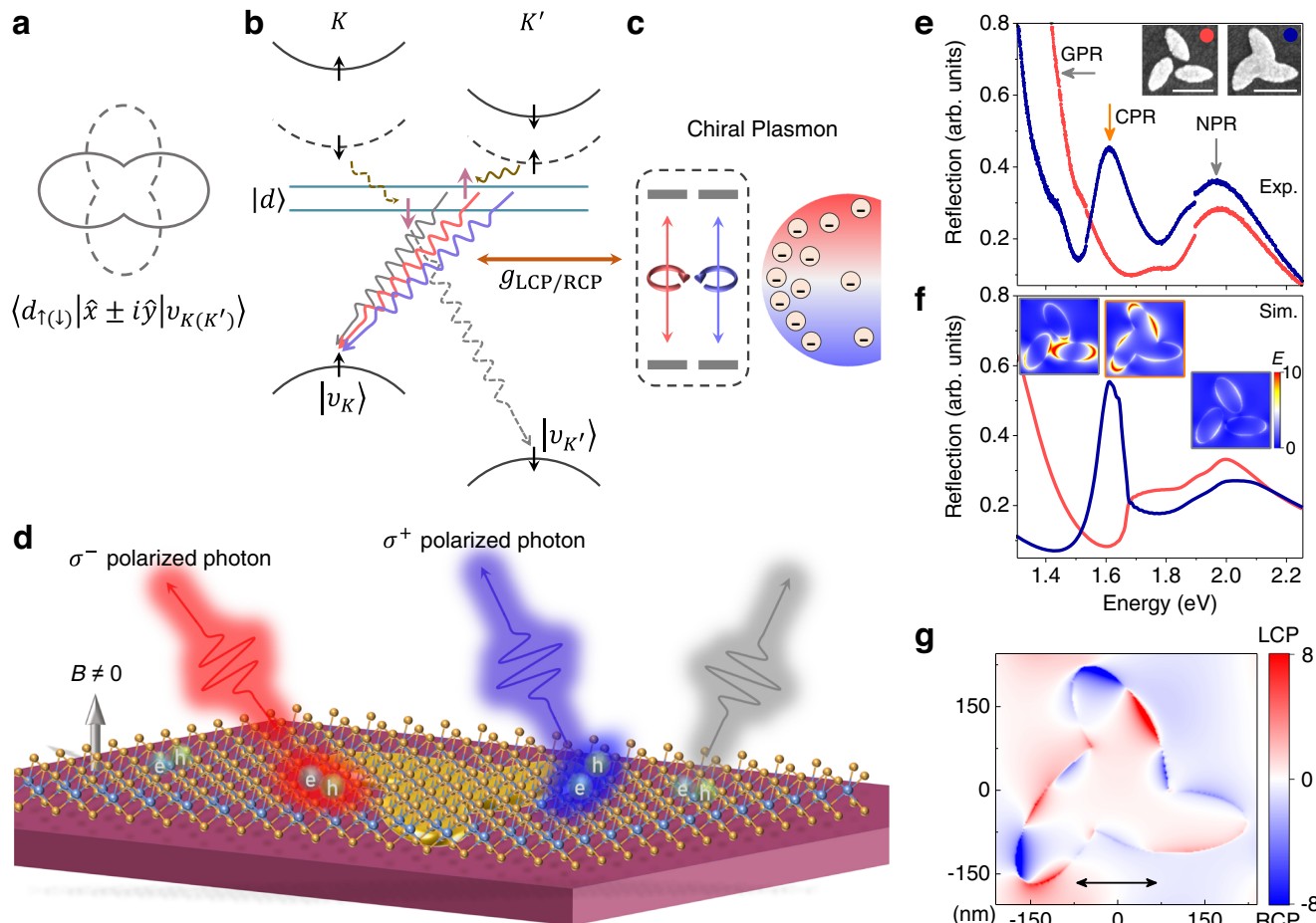

**Fig. 1 | Engineering of polarization states of quantum emitter (QE) with broken valley symmetry by chiral plasmonic nanocavities. a** The gray lines (8 shape) represent the emission of transitions $|d_{\uparrow(\downarrow)}\rangle \rightarrow |v_{K(K')}\rangle$ in a circular basis measurement at a large magnetic field, showing incomplete circularly polarized photon output in each channel. The $\hat{x}$ and $\hat{y}$ represents the displacement operators in the coordinate representation. **b** Schematic diagram of intervalley defect exciton in WSe₂ and the manipulation of circular polarization output by chiral plasmon field. The black curves show the valleys of *K* and *K'* shifted at a finite vertical magnetic field, where black arrows represent the spin. The hybridization of dark bands (black dashed curves) with defect levels (straight cyan lines) gives rise to electrons (pink arrows) independent of valley symmetry. The splitting of the defect level facilitates the hybridization of the lowest dark band and highest energy level due to the small energy difference and the same spin, which leads to the stronger low energy peak radiation of QE at a magnetic field. The wavy lines colored gray, red, and blue represent the bare QE emission and the emission when coupled with the left

circularly polarized (LCP) and right circularly polarized (RCP) components of the chiral plasmon field respectively. **c** Schematic diagram of chiral plasmon resonance which includes LCP (red color) and RCP (blue color) components. **d** Illustration of a cavity-dependent polarized photon output for QEs in WSe₂ with different chiral plasmon fields at a finite vertical magnetic field *B*. **e** The experimental (Exp.) results of the reflection spectra of plasmonic lattices, where the unit cell is formed by gold nanorods with a $C_3$ symmetry with corresponding scanning electronic microscopy (SEM) images shown in the insets. Scale bar is 250 nm. When the nanorods stick together, strong plasmon resonance is clearly shown at 1.61 eV. The arrows indicate the resonance modes of plasmonic lattices, including gap plasmon resonance (GPR), chiral plasmon resonance (CPR), and nanorod plasmon resonance (NPR). **f** The corresponding simulated (Sim.) reflection spectra and distribution of electric field intensity *E* of GPR, CPR, and NPR. **g** The calculated optical chirality enhancement of CPR. The black arrow indicates the linear polarization direction of excitation light.

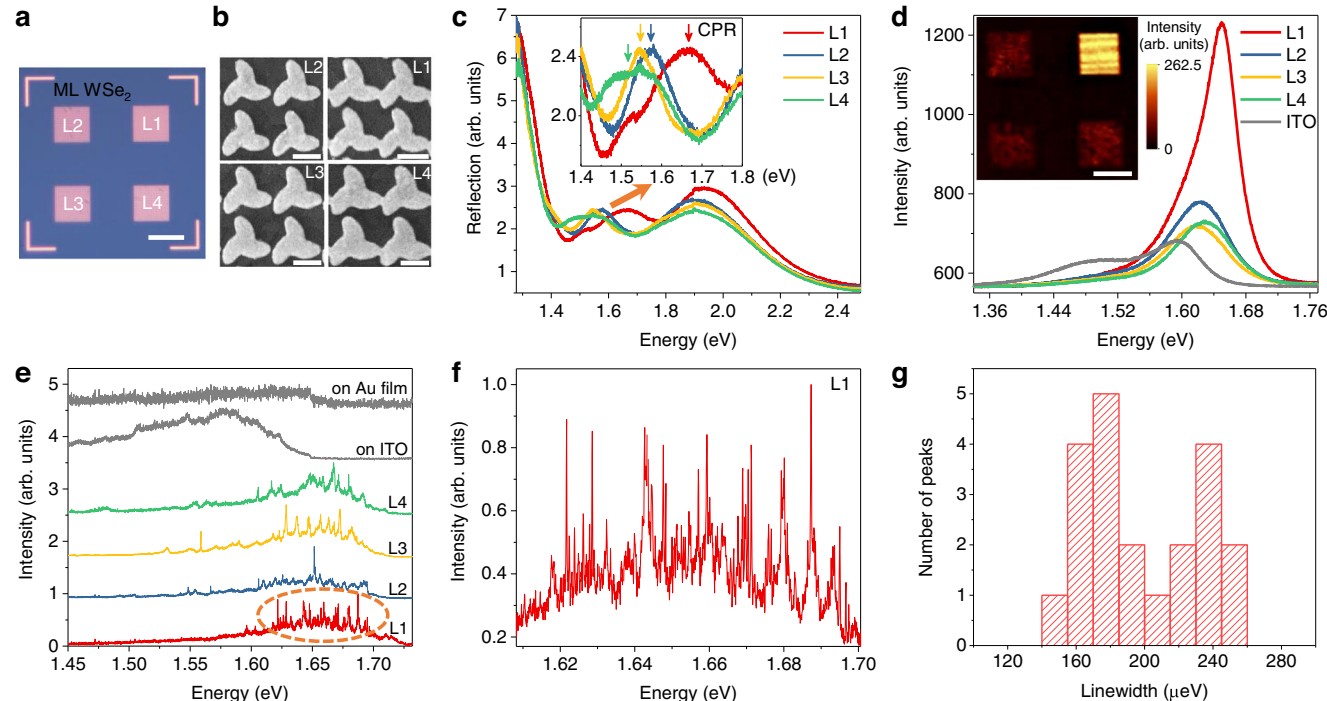

**Fig. 2 | Coupling between chiral plasmon resonance and single quantum emitters. a** Optical microscope image of monolayer (ML) WSe₂ covered on the plasmonic lattices L1, 2, 3, 4 with different periods and sizes. Scale bar, 10 µm. **b** SEM images of plasmonic lattices in **a**. Scale bar, 250 nm. Here, the period of L1 is 400 nm and the period of L2, 3, 4 is 450 nm. **c** Measured reflection spectra of plasmonic lattices. The inset shows enlarged CPR of L1, 2, 3, 4. **d** The photoluminescence (PL) spectra of the hybrid structures and monolayer on indium tin oxide (ITO) substrate at room temperature. Inset, corresponding PL mapping at 1.65 eV with an intensive PL enhancement at L1 due to the resonance of excitons with plasmon mode. Scale bar, 10 µm. **e** Low-temperature (4.2 K) PL spectra of monolayer flake on lattice structures and substrates. **f** Enlarged PL spectrum with massive sharp emission lines at L1 region. **g** The linewidth statistical histogram of the different sharp peaks with values less than 260 µeV, indicating the formation of quantum emitters.

offer an effective way to investigate those issues, because it can not only realize a resonant interaction with excitons and read out the information of a QE itself[31,32], but also bring opportunities for the research of cavity quantum electrodynamics and the exploration of new quantum optics devices[33,34]. Especially, plasmonic nanocavities have the ability to localize the electric field at nanoscale with localized surface plasmon polaritons[35,36], which can enhance the interaction with QEs greatly[13,14,35,37–39].

In this work by utilizing chiral plasmonic nanocavities, we experimentally observed that the polarization states of emitted photons from two different Zeeman splitting peaks are the same for the QEs coupled with chiral plasmonic nanocavities, suggesting the degree of circular polarization of the coupled QEs is independent of the transition channels from two different valleys, which reveals that the intrinsic valley protection is absent for those QEs in WSe₂. Moreover, our work develops a technique for control of light-matter interaction at the level of single quanta, indicating potential applications in chiral quantum optics[33], such as optical switches based on single chiral photons[40–42] and directional spin output of the intervalley defect excitons when embedded in nano-photonics devices[33,43–45].

## Results

### Single quantum emitters on chiral plasmonic lattices
Experimentally, the chirality of the plasmonic nanocavity is obtained by a periodic lattice arranged in squares, in which the unit cell of the lattice is formed by gold nanorods with a $C_3$ symmetry, as shown in the inset in Fig. 1e. When the nanorods stick together, an extra strong resonance peak arises in the reflection spectra with a narrow linewidth ~144 meV at 1.61 eV as shown in Fig. 1e, while only individual nanorod plasmon resonance (NPR) and gap plasmon resonance (GPR) can be seen for separated gold nanorods. Numerical simulation shows that

the quality factor of this mode is 5.3 times higher than the GPR (see Supplementary Fig. 5), and the field distribution of the plasmon mode is more localized around the edge of the chiral nanostructure as shown in Fig. 1f, suggesting a unique chirality of electromagnetic fields different from separated gold nanorods. Calculated optical chirality enhancement of the chiral plasmon resonance (CPR) is shown in Fig. 1g. It shows that both left circularly polarized (LCP) and right circularly polarized (RCP) components are strongly localized along the outer edge of the unit cell with a scale of tens of nanometers, comparable to the extending of localized QE excitons[11,12]. The local optical chirality can be enhanced by 8-14 at the hot spots than that without the lattice. Another advantage of this field distribution is that strain is more likely to occur at the edge of the metal structure, which facilitates the generation of QE and interacting with it.

After being patterned on the substrate, the chiral lattices were then covered by a WSe₂ monolayer, as shown in Fig. 2a. A large and flat monolayer is successfully obtained utilizing our exfoliation and transfer methods, as described in Methods. To match the energy of excitons, lattices L1 to L4 are designed with different parameters for desired plasmon resonance as shown in Fig. 2b. The adhesion between unit cells in L1 and L4 does not affect the CPR and chirality, which is confirmed by numerical simulations (see Supplementary Fig. 5). Figure 2c shows the reflection spectra of lattices with a CPR in the range of 1.45 to 1.75 eV, covering the energy range of valley excitons and defects in WSe₂. The CPR peaks shift with different structures (L1-L4) as designed. Figure 2d shows the photoluminescence (PL) spectra of the hybrid structures at room temperature. Clearly, an intensive enhancement at L1 can be observed due to the resonance between CPR and WSe₂ excitons. At low temperature, the PL spectra are dominated by defect-related recombination as shown in Fig. 2e. It is notable that the emissions from valley neutral excitons and trions are almost

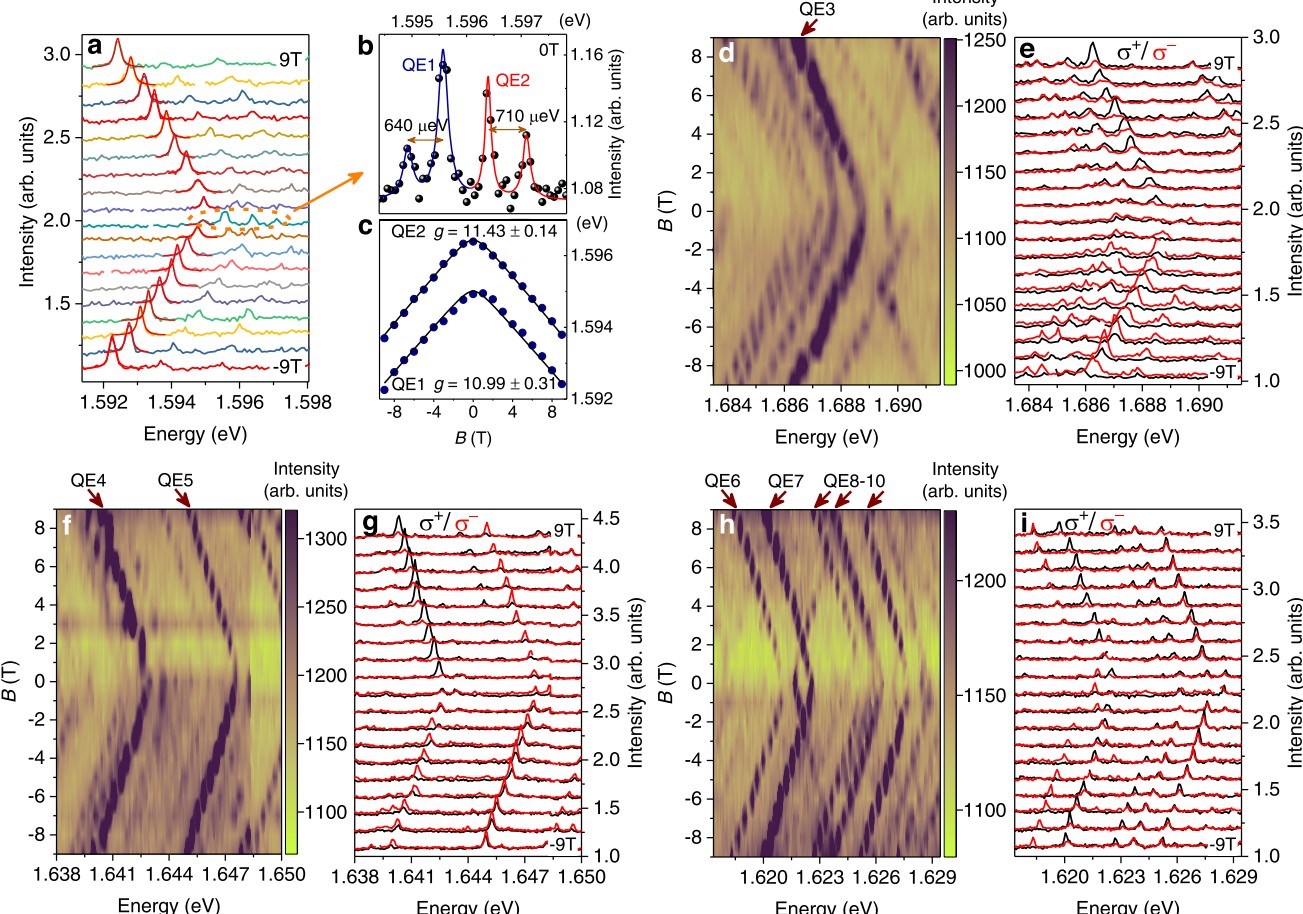

**Fig. 3 | Polarization-resolved PL from QEs at L1 as a function of applied magnetic field (Faraday geometry). a** PL spectra of QE1 and QE2 as a function of $B$, showing the splitting of two peaks with increasing $B$. The energy shifts of defect level and dark conduction band facilitate the hybridization process thus leading to stronger low energy peak radiation of QE. **b** The PL spectra measured at zero field, showing the fine structure splitting of active neutral QEs. **c** The fitted $g$-factors of QE1 and QE2. The peak data is obtained by fitting the peaks with Lorentz lineshape, as shown in the red lines in **a**. The larger $g$-factors of QEs stem from the net valence band shift when applying an out-of-plane magnetic field. **d**, **f**, **h** Measured magnetic field-dependent PL of QE3-QE10, no polarizer or wave plate was inserted during the collection. **e**, **g**, **i** Polarization-resolved magnetic field dependent PL of QE3-QE10 which was detected in the circular basis (black lines: $\sigma^+$ configuration; red lines: $\sigma^-$ configuration). All the measurements were performed with a linearly polarized excitation. The spectra are shifted for clarity.

quenched in our sample. This is because the nonradiative energy transfer between the Au or ITO substrate and long-lived low-energy dark states increases the relaxation rate of the dark state in $WSe_2$, thus reducing the population of both dark and bright exciton states by phonon scattering between two states. However, the energy of defect states is lower than that of dark states of conduction band. As a result, the defect states do not experience this quenching effect[46]. Different from the PL spectra on ITO and Au film with a broad defect ensemble peak, PL spectra at L1-L4 present sharp peaks superimposed on the relatively broad defect emission peak with a linewidth of around hundreds of µeV, which come from atom-like QEs[3,14,37]. Especially at L1, massive sharp emission lines can be resolved in PL spectrum (as enlarged in Fig. 2f). The energies of the sharp lines are in the range of 1.62–1.70 eV, which are in resonance with CPR at L1, indicating a coupling between the plasmon and quantum emitters. The statistical histogram of the linewidths for 21 peaks is presented in Fig. 2g, with a range from -148 to 256 µeV, similar to the values of single quantum emitters reported previously[1–4].

## Polarization-resolved magneto-photoluminescence spectroscopy of QEs

To investigate the polarization states of QEs, lifting of valley degeneracy and overcoming the electron-hole exchange interaction are necessary.

Figure 3a shows the PL spectra of QEs as a function of applied out-of-plane magnetic field $B$. At zero field, the QEs exhibit an obvious fine structure splitting, and the energy splitting of the doublets is about 600-700 µeV as shown in Fig. 3b, indicating optically active neutral QEs in our sample[1,3,4,24], which is also confirmed by the value of the $g$-factors (-11.0, see in Fig. 3c and see also Supplementary Fig. 7). As the magnetic field increases, the intensity of the low energy peak of Zeeman splitting increases gradually while the high energy peak decays rapidly and almost disappears at high magnetic fields. That was previously attributed to the thermal relaxation[1,3,24,47], but it is insufficient. As shown in Fig. 1b, the energy of the defect level of the low energy peak is actually higher than that of the high energy peak due to the large shift of valence band under magnetic field. This would typically lead to a lower probability of electron distribution in the defect level of the low energy peak, as predicted by Fermi-Dirac statistics. However, our experiments show the opposite results, indicating that the low energy peak does not decay as expected. Here, we ascribe the magnetic field-dependent PL intensity change to the hybridization of the defect level with dark conduction band. As shown in Fig. 1b, the defect level state $|d_\uparrow\rangle$ with spin up will move up while the lowest dark band state $|c_{K'}\rangle$ with the same spin will move down when applying a positive magnetic field. The decrease of the energy difference between defect level and conduction band will facilitate the hybridization process similar to that caused by strain[9,29],

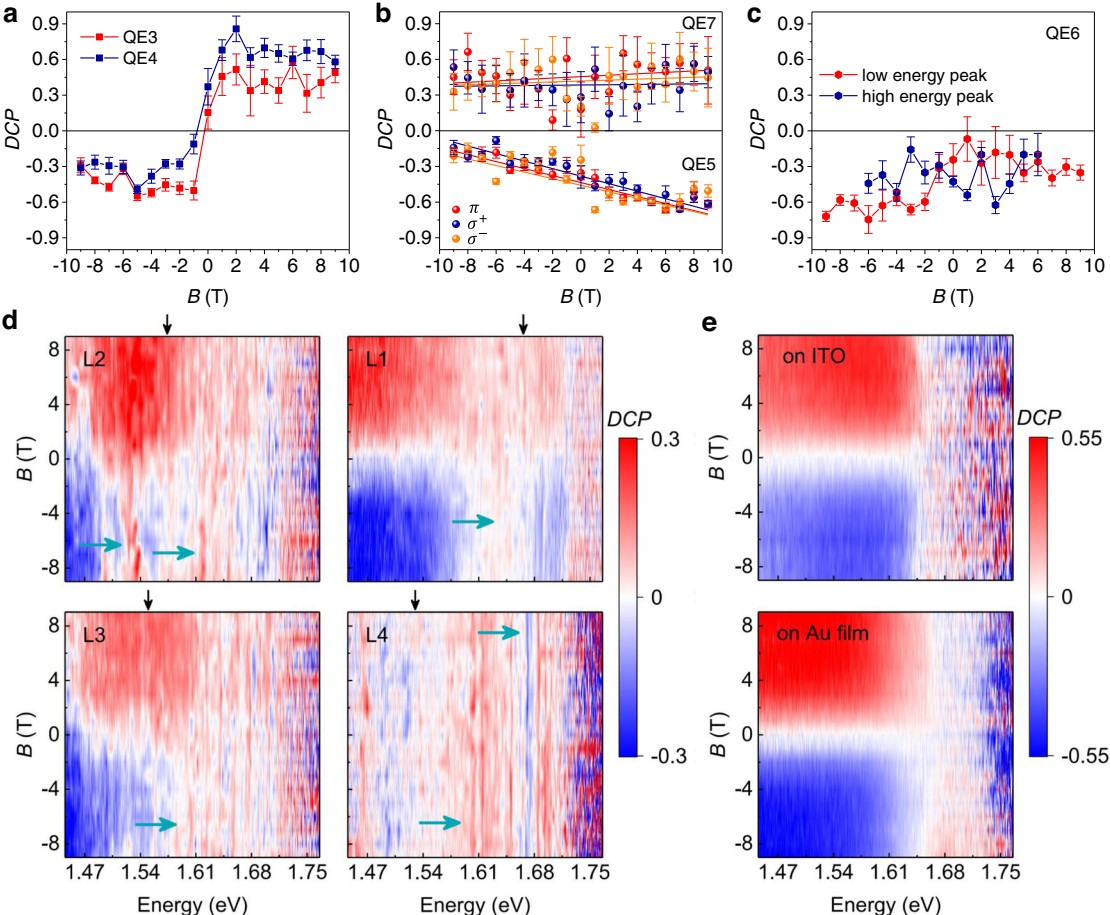

**Fig. 4 | Measured magnetic field-dependent degree of circular polarization (DCP), $(I_{\sigma^+} - I_{\sigma^-})/(I_{\sigma^+} + I_{\sigma^-})$, where $I_{\sigma^+}$ ($I_{\sigma^-}$) denotes the intensity of the $\sigma^+$ ($\sigma^-$) polarized emission. a** DCP of QE3 and QE4, showing a flip of helicity with the magnetic field. **b** DCP of QE5 and QE7 under linearly ($\pi$), $\sigma^+$ and $\sigma^-$ polarized excitation and corresponding linear fitting (lines), showing constant helicities after experiencing an applied magnetic field of −9 T to 9 T. No observable difference can be seen with different excitation, suggesting the absence of valley polarization. The DCP of QE5 is in the range of −17% ~−69% and QE7 is 38% ~45%. **c** Both the low energy

peaks and high energy peaks show the same chirality output from QE6. The high-energy branch disappears when $|B| > 6$ T. The source of error bars is the standard error of QE intensity in peak fitting. **d** DCP of L1, 2, 3, 4 with high power excitation to obtain a collective behaviour of massive defects. The cyan arrows indicate the emissions with the same helicities. The black arrows mark the energy peaks of the reflection spectra of lattices. **e** Measured DCP on ITO and Au film for a comparison with **d**, showing an obvious flip of helicity with opposite magnetic fields.

resulting in stronger $|d_\uparrow\rangle \rightarrow |v_K\rangle$ radiation i.e. low energy peak emission. When applying a negative magnetic field, the $|d_\downarrow\rangle \rightarrow |v_{K'}\rangle$ radiation will become a low energy peak and the corresponding emission will be promoted by the magnetic field. As for the high energy peak, the energy difference between the defect level and the dark conduction band with the same spin becomes larger under magnetic field, leading to a rapid weakening of hybridization. Thus almost only low energy PL peak appears in a magnetic field, which further proves that the intervalley defect excitons are the sources for quantum emission in WSe₂.

In addition, output polarization states of those neutral intervalley defect excitons could be controlled by the external chiral plasmon. Figure 3d−i show the polarization-resolved magnetic field-dependent PL spectra. Surprisingly, QE5, QE6, and QE7 show almost the same helicities with an applied magnetic field from −9 T to 9 T, which is different from the QEs in previous studies[1,47]. Previously, a reversal of the circular polarization of low energy peak was demonstrated when an opposite magnetic field was applied, similar to the properties of QE3 and QE4 in our sample. Figure 4a shows the degree of circular polarization (DCP) of QE3 and QE4, where the low energy peak is $\sigma^+$ polarized at positive $B$ and $\sigma^-$ polarized at negative $B$ due to the transitions from two different channels with opposite spins. The value of DCP for QE3 reaches ~± 50% and becomes saturated when the $|B| \geq 1$ T. This is because the exchange energy is took over by the Zeeman

interaction and circularly polarized transitions are recovered[48]. At $B = 0$, the DCP is small but not equal to zero, which may be due to measurement error caused by the weak signal at small $B$, as shown in Fig. 3d−i. Similar behaviors can be found in QE4, except that the saturation values of DCP are asymmetric, ~−30% at a negative magnetic field and ~60% at a positive field. Different from QE3 and QE4, QE5 and QE7 show an anomalous circularly polarized emission. Either $\sigma^-$ polarized or $\sigma^+$ polarized output is only observed whenever at positive or negative magnetic fields, as shown in Fig. 4b. The DCP measured under different polarized excitation shows no difference with linear fitting, which is quite different from the valley excitons with ~30% valley polarization (see Supplementary Fig. 10).

## Cavity-dependent circularly polarized single-photon output

We attribute this anomalous circular polarization to two reasons. One is the absence of valley polarization during QE transition, and the other is the coupling with the chiral plasmon field, resulting in a circularly polarized photon output independent of the transition channels of the two different valleys ($|d_{\uparrow(\downarrow)}\rangle \rightarrow |v_{K(K')}\rangle$). This phenomenon is quite different from valley excitons, in which the transition is determined by the valley-dependent optical selection rules, and the decrease in DCP is caused by relaxation processes, such as phonon-assisted intervalley scattering[49,50]. The presence of CPR may strengthen or weaken the

relaxation process, leading to a further reduction or increase in the *DCP*, but it will not be reversed. For example, the scattering of valley excitons can be reduced by the microcavity[51,52], but the valley polarization of a certain valley cannot be reversed. In our experiments, we observed a surprising reversal of polarization in one transition channel, that is, the nanocavity-dependent circularly polarized photon output, confirming the unique property of QEs that are different from valley excitons and proving the absence of valley polarization for the intervalley defect exciton (more discussions in Supplementary Fig. 2). Results with polarized excitations of the QEs coupled with chiral nanocavities further confirm the absence of valley polarization, indicating the breaking of valley symmetry in WSe₂ QEs. The *DCP* of QE5 changes a little with the magnetic field, which may be due to coupling strength change with the applied magnetic fields[53]. The overlap of the exciton wave function with the plasmon hot spots could be tuned by the magnetic field, especially when the location of the defect is at the boundary of the chiral field region. QE6 gives another evidence of the coupling with plasmonic nanocavity and the breaking of valley symmetry since the same circularly polarized low energy and high energy PL peaks are observed, as shown in Fig. 4c. We performed statistics on the coupled QEs with anomalous behaviour, that is, QEs with the same helicities under opposite magnetic fields (see Supplementary Fig. 13). We found that about half of QEs at L1 (14/29) are coupled to CPR. This relatively high proportion can be attributed to the fact that the chiral plasmon field is predominantly concentrated at the edge, where the QEs are more likely to be generated due to the strain.

To further demonstrate the control of polarization states by chiral fields is universal, Figure 4d shows the *DCP* maps of defects on L1-L4 under a high-power excitation, with which emission from massive defects can be observed. For comparison, the emission of defects in the WSe₂ layer on ITO substrate and Au film are shown in Fig. 4e, where a high *DCP*-50% can be observed and flipped when the magnetic field reversed. Emissions from defects on plasmonic lattices L1 to L4 show a lower *DCP*. In some energy range, the helicity does not flip with the magnetic field, as marked by the cyan arrows in Fig. 4d. The location of these anomalous peaks is related to the energy of CPR and the uneven distribution of defects. This anomalous radiation behaviour displayed by a large number of QEs indicates that our observations are not coincidental. Overall, more σ⁺ emission can be observed, which is due to that the unit cells of lattices rotate in the same direction, causing this imbalance in circular outputs. This abnormal circular output of quantum emitters comes from the fact that the intervalley defect exciton is coupled to the chiral localized plasmon field. As illustrated in Fig. 1a, the uncoupled transitions of QE are not perfectly circularly polarized for the two different channels $|d_{\uparrow(\downarrow)}\rangle \rightarrow |v_{K(K')}\rangle$ between the defect states and two valence bands, which is also confirmed by the measured *DCP* as shown in Fig. 4a, e. Even after overcoming the electron-hole exchange interaction at a higher magnetic field, the saturated *DCP* around 50% is still much lower that in In(Ga)As/(Al)GaAs quantum dots with a complete circular polarization[44,48,54]. Previous experiments[24,47] and density functional theory (DFT) calculations[29] also show similar results that a partial circular polarization with cross-circularly polarized components mixed in the emission spectra of WSe₂ QEs. Thus, we can consider that the QE itself doesn't obey the selection rules strictly as the exciton angular momentum of QE is not a good quantum number. Since the hole is a single particle state in the valence band, it is different from the energy levels of traditional quantum dots. Therefore, the transition matrix element $\langle d_{\uparrow(\downarrow)} | \hat{x} \pm i\hat{y} | v_{K(K')}\rangle$ for σ⁺ polarization will contain two components of cross-circular polarization for each transition channel. The CPR can not only enhance the radiation of QEs, but also facilitates the co-polarized absorption process. This offers an opportunity to manipulate the polarization of output photons using the chiral plasmon field.

Similar experiments on MoSe₂ monolayer verify that this cavity-dependent chiral photon output is unique to the intervalley defect excitons (see Supplementary Fig. 18). The valley exciton in MoSe₂ is "bright" since optical intravalley transitions between valence band maximum and conduction band minimum are spin allowed. Unlike in WSe₂, the defect states in MoSe₂ do not need to undergo cross-valley hybridization to produce a strong photons emission. Thus, the defect excitons in MoSe₂ monolayer are intravalley defect excitons and the theoretical calculations indicate that the transitions between valence band and defect levels have the same optical selection rules as the valley exciton at $K$ and $K'$ points[27]. The measured magnetic field-dependent *DCP* of MoSe₂ monolayer on chiral plasmonic lattices shows the circularly polarized emission of defect states will no longer be modified by chiral plasmon field. Therefore, a reversal of *DCP* that happens in WSe₂ will not be observed in MoSe₂ due to the valley-dependent selection rules. It only shows a modification of circularly polarized emission in the far-field by the chiral plasmon resonance at lower magnetic fields.

## Dynamics of coupled QE-CPR system

To investigate the underlying mechanism of the anomalous chiral photon emission from the WSe₂ monolayer, we employed a dynamical solving approach to study the coupled QE-CPR system. Figure 5a sketches the coupling between the intervalley defect exciton and chiral plasmonic nanocavity. The system can be described by the Hamiltonian $H = H_{QE} + H_{pl} + H_{pump} + H_{int-\sigma^+} + H_{int-\sigma^-}$, where

$$H_{QE} + H_{pl} = -\hbar\omega_{cd}\left|d_{\uparrow(\downarrow)}\right\rangle\left\langle d_{\uparrow(\downarrow)}\right| - (\hbar\omega_{QE} + \hbar\omega_{cd})\left|v_{K(K')}\right\rangle\left\langle v_{K(K')}\right|$$
$$+ \hbar\omega_{pl}(\hat{a}_{\sigma^+}^\dagger\hat{a}_{\sigma^+} + \hat{a}_{\sigma^-}^\dagger\hat{a}_{\sigma^-}) \quad (1)$$

$$H_{pump} = \Omega\left(e^{i\hbar\omega_L t}\left|v_{K(K')}\right\rangle\left\langle c_{K'(K)}\right| + e^{-i\hbar\omega_L t}\left|c_{K'(K)}\right\rangle\left\langle v_{K(K')}\right|\right) \quad (2)$$

$$H_{int-\sigma^\pm} = g_{\sigma^\pm}\left(\left|d_{\uparrow(\downarrow)}\right\rangle\left\langle v_{K(K')}\right|\hat{a}_{\sigma^\pm} + \left|v_{K(K')}\right\rangle\left\langle d_{\uparrow(\downarrow)}\right|\hat{a}_{\sigma^\pm}^\dagger\right) \quad (3)$$

Here, $H_{QE}$ and $H_{pl}$ describe the QE states and the chiral plasmon mode, respectively. The state $|c_{K'(K)}\rangle$ is the lowest dark conduction band and the corresponding energy is set to zero. $\hbar\omega_{QE}$, $\hbar\omega_{cd}$ represent the energy of QE and the energy difference between dark conduction band and defect level, respectively. The quantized chiral plasmon mode $H_{pl}$ with resonance energy $\hbar\omega_{pl}$ is described by circularly polarized operators $\hat{a}_{\sigma^\pm} = \frac{1}{\sqrt{2}}(\hat{a}_1 \pm i\hat{a}_2)$, which is satisfying $[\hat{a}_{\sigma^\pm}, \hat{a}_{\sigma^\pm}^\dagger] = 1$[55]. Here, we assume that the external pump laser field $\varepsilon = \Omega e^{i\hbar\omega_L t}$ brings the exciton states into the excited intervalley dark exciton ($I_D$) states as described by $H_{pump}$. This intervalley dark exciton will decay to the optically active defect state $|d_{\uparrow(\downarrow)}\rangle$ by nonradiative hybridization. Since the photon from low energy branch has both σ⁺ and σ⁻ polarized components, the interaction Hamiltonian $H_{int}$ consists of two parts. One is coupled to σ⁺ and the other is coupled to σ⁻ polarized component of plasmon field. The ratio of coupling strength $g_{\sigma^+}/g_{\sigma^-}$ will depend on the location of QE at LCP or RCP chiral hot spots. The dynamics of QE-CPR system can be obtained by solving the time dependent quantum master equation, which is $\dot{\rho} = -i[H,\rho] + \sum_n \hat{C}_n\rho\hat{C}_n^\dagger - \sum_n \frac{1}{2}(\hat{C}_n^\dagger\hat{C}_n\rho + \rho\hat{C}_n^\dagger\hat{C}_n)$ in Lindblad form[56,57], where $\rho$ is the reduced density matrix of the system. The items containing $\hat{C}_n$ describe the non-hermitian evolution of the system due to its coupling to the environment, which is responsible for irreversible dissipation[58], including QE decay $\hat{C}_{QE} = \sqrt{\gamma_{QE}}|v_{K(K')}\rangle\langle d_{\uparrow(\downarrow)}|$ with rate $\gamma_{QE}$, nonradiative hybridization $\hat{C}_{hyb} = \sqrt{\gamma_{hyb}}|d_{\uparrow(\downarrow)}\rangle\langle c_{K'(K)}|$ with strength $\gamma_{hyb}$ and chiral plasmon decay $\hat{C}_{pl_{\sigma^\pm}} = \sqrt{\gamma_{pl}}\hat{a}_{\sigma^\pm}$ with rate $\gamma_{pl}$.

The radiation power of coupled QE-CPR system can be obtained by solving the steady solution of the master equation. We first focus on

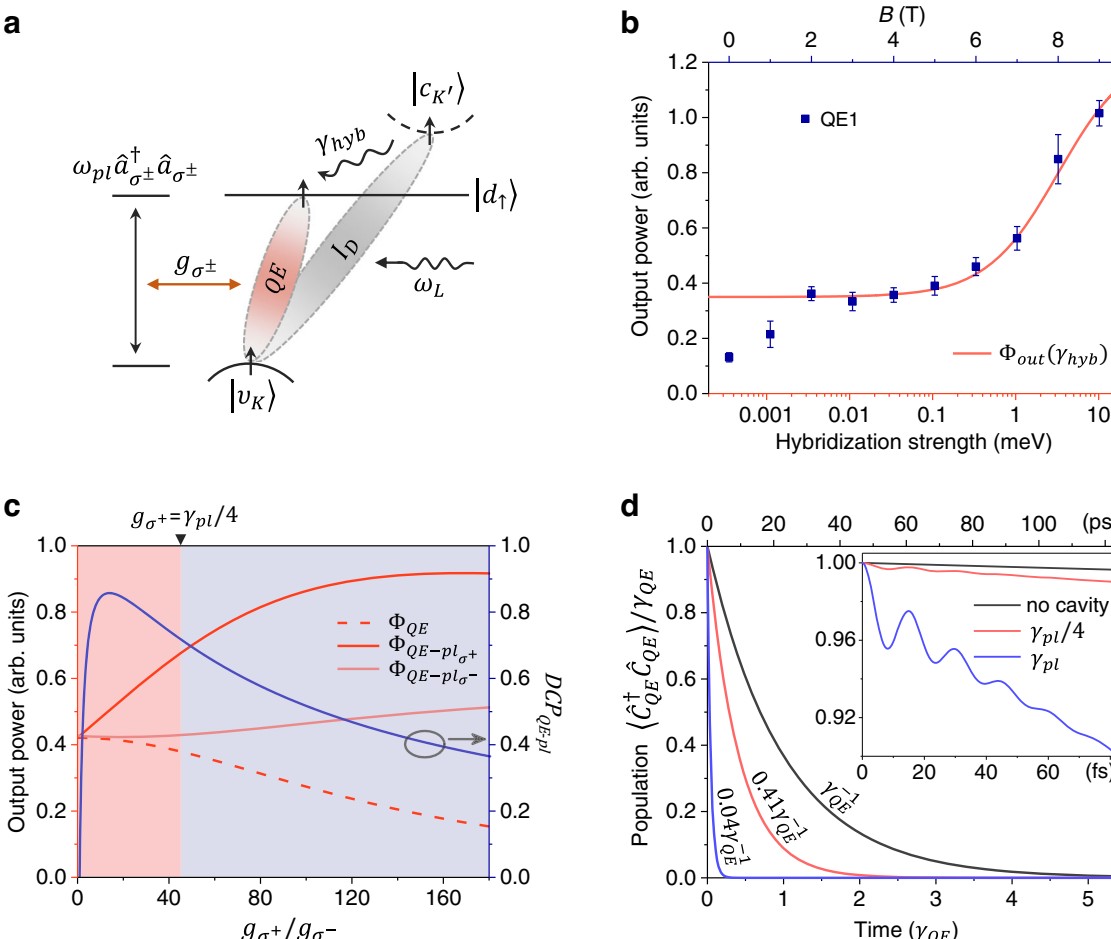

**Fig. 5 | Dynamics of coupled QE-CPR system. a** Schematic illustration of QE-CPR interaction. The external pump laser with frequency $\omega_L$ brings the exciton states into the excited intervalley dark exciton ($I_D$) states. Then the intervalley dark exciton decays from the lowest dark conduction band $|c_{K'}\rangle$ to the optically active defect state $|d_\uparrow\rangle$ by nonradiative hybridization with strength $\gamma_{hyb}$. The transition of QE between $|d_\uparrow\rangle$ and $|v_K\rangle$ can couple to the chiral plasmon mode $\hbar\omega_{pl}\hat{a}^\dagger_{\sigma^+}\hat{a}_{\sigma^+}$ and $\hbar\omega_{pl}\hat{a}^\dagger_{\sigma^-}\hat{a}_{\sigma^-}$ with coupling strength $g_{\sigma^+}$ and $g_{\sigma^-}$, respectively. **b** The output power of QE as a function of magnetic field and hybridization strength. The red curve is calculated by master equation and the blue dots are taken from the experimental results of QE1 as shown in Fig. 3a. The error bar is derived from the standard error of QE intensity in peak fitting. The dots below 1 T do not fit well with the curve due to

the exchange interaction. **c** Power outputs of different channels of coupled QE-CPR system versus the chiral coupling strength ratio: radiation of QE $\Phi_{QE} = \langle\hat{C}^\dagger_{QE}\hat{C}_{QE}\rangle$, radiation of coupled QE and $\sigma^+$ polarized plasmon $\Phi_{QE-pl_{\sigma^+}} = \langle\hat{C}^\dagger_{QE-pl_{\sigma^+}}\hat{C}_{QE-pl_{\sigma^+}}\rangle$, radiation of coupled QE and $\sigma^-$ polarized plasmon $\Phi_{QE-pl_{\sigma^-}} = \langle\hat{C}^\dagger_{QE-pl_{\sigma^-}}\hat{C}_{QE-pl_{\sigma^-}}\rangle$. The blue line represents the calculated $DCP_{QE-pl}$. The red and blue regions represent the weak and strong coupling regimes, respectively. **d** Temporal evolution of the population $\langle\hat{C}^\dagger_{QE}\hat{C}_{QE}\rangle/\gamma_{QE}$ on the QE with coupling strength $g_{\sigma^+} = 0, \gamma_{pl}/4$ and $\gamma_{pl}$, respectively. The black line shows the population without cavity. The inset shows the QE population on femtosecond timescale. The pump term is removed in the calculation in order to observe the time evolution of the coupled system. The initial state is QE in the excited state.

the effect of hybridization strength on the output power of QE. When the coupling between QE and plasmon is very weak, the output of the coupled system can be expressed as $\Phi_{out} = \Phi_{QE} + \Phi_{pl}$. $\Phi_{pl}$ represents the radiation of plasmon which is given by $\Phi_{pl} = \langle\hat{C}^\dagger_{pl}\hat{C}_{pl}\rangle$, where $\hat{C}_{pl} = \sqrt{\gamma^r_{pl}}(\hat{a}_{\sigma^+} + \hat{a}_{\sigma^-})$ and $\gamma^r_{pl}$ is the decay rate of radiation from plasmonic nanocavity to vacuum. The blue dots in Fig. 5b show the experimental output power of QE as a function of magnetic field. Because the energy difference between dark band and defect level is linear to the magnetic field, and directly related to the hybridization strength, we can fit the measured data with the calculated output $\Phi_{out}$ to show the scale of nonlinear hybridization under linear energy difference, as shown by the red line in Fig. 5b. We find that the experimental results fit well with the calculated results when the hybridization strength increases by about 4 orders of magnitude as the energy approaches. It confirms that the enhancement of low energy

peak and the disappearance of high energy peak with $B$ stem from the nonlinear hybridization with the lowest dark conduction band. This PL enhancement and nonlinear hybridization between defect level and dark conduction band have also been observed in mechanical strain engineering in WSe$_2$ monolayer[30].

For the coupled QE-CPR system, the output power is given by $\Phi_{QE-pl_{\sigma^\pm}} = \langle\hat{C}^\dagger_{QE-pl_{\sigma^\pm}}\hat{C}_{QE-pl_{\sigma^\pm}}\rangle$ for the couplings with $\sigma^\pm$ polarized components of chiral plasmon mode. Here, $\hat{C}_{QE-pl_{\sigma^\pm}} = \sqrt{\gamma_{QE}}|v_{K(K')}\rangle\langle d_{\uparrow(\downarrow)}| + \sqrt{\gamma^r_{pl}}\hat{a}_{\sigma^\pm}$ represents the radiation to vacuum from both QE and chiral plasmon mode[59,60]. Figure 5c shows the power outputs of different channels in the coupled system as a function of the chiral coupling strength ratio when the QE is located at chiral hot spots with $\sigma^+$ polarized plasmon field dominated. As $g_{\sigma^+}$ increases (take $g_{\sigma^-} = 1$ meV), the radiation from QE $\Phi_{QE}$ gradually decreases while the output power of the coupled QE-nanocavity system increases. The

radiation from coupling with $\sigma^+$ polarized plasmon $\Phi_{QE-pl_{\sigma^+}}$ shows a stronger intensity compared to that of $\sigma^-$ polarized plasmon $\Phi_{QE-pl_{\sigma^-}}$. The output of $\Phi_{QE-pl_{\sigma^+}}$ does not increase all the time, but saturates in strong coupling regime $(2g_{\sigma^+} > (\gamma_{QE} + \gamma_{pl})/2 \approx \gamma_{pl}/2)$ as shown in the blue region in Fig. 5c. Here, the radiation of coupled system without the QE part, that is $\Phi_{QE-pl_{\sigma^\pm}} - \Phi_{QE}$, is known to be circularly polarized (see details in Supplementary Note 1). The degree of circular polarization can be calculated with $DCP_{QE-pl} = (\Phi_{QE-pl_{\sigma^+}} - \Phi_{QE-pl_{\sigma^-}})/(\Phi_{QE-pl_{\sigma^+}} + \Phi_{QE-pl_{\sigma^-}} - 2\Phi_{QE})$, as the blue line shown in Fig. 5c. At the coupling strength ratio of ~14.1, $DCP_{QE-pl}$ has a maximum value of about 85%, and then it decays to less than 40% as the coupling strength increases. Although the output proportion of the nanocavity part is higher than that of QE in the strong coupling regime, circular polarization degree of output is still not high. In the weak coupling regime, the output proportion of QE is relatively high, so a large part of the circular polarization output is contributed by the QE. Utilizing the $DCP_{QE-pl}$, we can estimate the contribution of QE to the overall experimentally measured $DCP$ by $DCP_{QE} = DCP + \beta(DCP - DCP_{QE-pl})$, where $\beta = (\Phi_{QE-pl_{\sigma^+}} + \Phi_{QE-pl_{\sigma^-}} - 2\Phi_{QE})/\Phi_{QE}$ (see Supplementary Note 1). For example for QE7 in weak coupling regime $(0 < g_{\sigma^+} < \gamma_{pl}/4)$, the $DCP_{QE}$ of ~21–45% can be obtained when the measured $DCP$ of 45% is used. This large contribution of circularly polarized output from quantum emitter indicates the spontaneous emission of QE is strongly modified by the chiral plasmon field. The value of $DCP_{QE}$ is constant under both positive and negative magnetic fields, that is, it is independent of different transition channels, implying the absence of the valley-dependent optical selection rules and the intrinsic valley protection. We can conclude that magnetic field-independent circular polarization output is dominated both by the QE and plasmonic nanocavity in our experiments.

Figure 5d shows the calculated time evolution of the emitter population. It can be seen that the lifetime of QE is largely reduced when coupling with plasmonic nanocavity. In the weak coupling regime, the lifetime of QE can be reduced to 40% comparing with that without the cavity. When the coupling strength $g_{\sigma^+}$ is stronger than the loss of plasmon, the lifetime can even be reduced by more than 25 times. The population on femtosecond timescale shows a rapid oscillation decay within about 60 fs whenever in weak coupling or strong coupling condition, indicating an ultrafast energy transfer between the QE and plasmon field as shown in the insert in Fig. 5d. For our system, the lifetime of deterministic circularly polarized single-photon output could be switched from picosecond with QE radiation to femtosecond QE-CPR radiation. This is crucial to realize novel plasmonic devices such as single-photon transistors[61], all-optical switches[62,63] and quantum information processors[64–66].

## Discussion

In summary, we show experimental evidence that the single quantum emitter in the two-dimensional WSe$_2$ monolayer breaks the valley symmetry by interacting with the chiral plasmonic nanocavity. The helicities of emitted photons remain consistent across two different Zeeman splitting peaks, suggesting the circularly polarized states of output photons are dominated by the chiral nanocavity instead of the valley-dependent optical selection rules. Our results also show that the intervalley defect exciton is one of the microscopic origins of the QEs in transition-metal dichalcogenides monolayer. Moreover, our work provides an experimental method to engineer the single-photon emission in desired circularly polarized states by chiral plasmonic nanocavities. The polarization state of the output single photon from the intervalley defect exciton can be switched by the chirality of the nanocavity. By locking the spin angular momentum of the output photon of an exciton state by chiral nanocavities, the system can be used to demonstrate many new photonics devices, such as single-photon sources with deterministic chirality, or chiral

single-photon switches when the system is optimized in a strongly coupled regime[40–42]. Directional output of one certain spin state at a high magnetic field can be achieved when the coupled system is embedded into nano-photonics devices such as waveguides or optical circulators based on the chiral coupling[33,43–45]. The $DCP$ of QEs steered by chiral plasmon in our work can reach up to ~60%, which is comparable to that generated by the Zeeman effect. Future exploration may lead to circularly polarized states with higher purity, providing an ideal platform for integrated single-photon-based plasmonic photonics.

## Methods
### Sample fabrication
The large-area WSe$_2$ monolayer was prepared by a gold-assisted mechanical exfoliation method, utilizing an Au adhesion layer with covalent-like quasi-bonding to layered crystal which provides access to a broad spectrum of large-area monolayer materials[67]. By using an electron beam evaporator, an adhesion metal layer (Ti or Cr) was first evaporated on SiO$_2$/Si substrate (with a SiO$_2$ thickness of ~300 nm), followed by Au film deposition on Ti or Cr layer with an evaporation rate of 0.5 Å/s. Then a fresh surface of WSe$_2$ crystal was cleaved from the tape and put it onto the Au film substrate. By pressing the tape vertically for about 1 min, the tape can be removed from the substrate. Large-area monolayer flakes can be easily obtained. One layer of Poly (methyl methacrylate) (PMMA) was then spin-coated on the exfoliated sample after exfoliation. Large-area WSe$_2$ together with PMMA was detached from the substrate after etching Au film away by KI/I$_2$ solvent. Then, the WSe$_2$-PMMA film was cleaned by water and picked up by the pre-patterned chiral plasmonic lattices utilizing flexible non-destructive wet transfer techniques. Finally, the sample was annealed at 110 °C for 3 minutes. To fabricate the chiral plasmonic lattices, a layer of PMMA (PMMA 495 K) was spin-coated on ITO glass substrates (ITO thickness 180 nm) with a thickness of ~100 nm. The designed chiral nanostructures were then patterned with electron beam lithography. After the development, metal layers with 5 nm Ti and 30 nm Au were evaporated using an electron beam evaporator. After the metal deposition, a lift-off procedure was performed to obtain chiral plasmonic nanostructures on the substrates. A 2 nm Al$_2$O$_3$ layer on top of the structures has been grown by atomic layer deposition as a plasmonic spacer.

### PL and reflection spectroscopy
The experiments of magneto-optical spectroscopy were performed in a helium bath cryostat. The sample was placed on a 3D piezo-electric stage and cooled down to 4.2 K via heat change with helium gas. The superconducting magnet surrounding the sample can supply a vertical magnetic field from −9 T to 9 T and horizontal magnetic field from −4 T to 4 T. PL spectroscopy was performed using confocal microscope set-up with a continuous laser of 532 nm for excitation. The emission was collected using a ×50 objective (numerical aperture 0.55) and directed to a grating spectrometer through optical fiber where a 1200 g/mm grating was used for high-resolution spectra. A liquid nitrogen-cooled charge-coupled device was used as a detector. Circular polarization-resolved measurements were performed by using a $\lambda/4$ plate followed by a polarizer. The Raman and room temperature PL measurements were operated on a WITec alpha300R system equipped with a pumping laser at 532 nm, and the scanning step of PL mappings was 500 nm per step.

The reflection spectra of plasmonic lattices were collected using a halogen lamp as excitation. The optical fiber is used to direct the unpolarized white light into the optical path. An objective (×100, numerical aperture 0.7) was able to focus the light spot to an area of tens of microns, comparable to the region of

plasmonic lattices. The reflection signal was collected by the same objective and the spectra were recorded by the same spectrometer as above with a 300 g/mm grating. The reference signal was collected from the position next to the area of interest but no structure. For measurement of circular dichroism (CD), a linear polarizer and a $\lambda/4$ were inserted into the input optical path at specific orientations. All the reflection spectra and CD were measured at room temperature.

## Numerical simulations

The numerical simulations using the finite-difference time-domain method were performed to calculate optical properties of the plasmonic lattices resonances. Near-field information including complex electric and magnetic field data and the far-field reflection are detected by field monitors with an excitation of a plane wave at normal incidence. The parameters of gold chiral structures were taken from SEM images and the permittivity of gold was taken from Johnson and Christy[68]. The refractive index of 1.65 was assumed for the ITO substrate to obtain spectral overlap with the experiments. The spatial grid sizes for gold structures were set to 2 nm in all directions. The complex electric and magnetic field data are taken from the plane 1 nm above the top surface of gold nanorods.

The optical chirality is defined as $C(\mathbf{r}) = -\frac{1}{2}\epsilon_0\omega \cdot \mathrm{Im}[\vec{\mathcal{E}}^*(\mathbf{r}) \cdot \vec{\mathcal{B}}(\mathbf{r})]$, where $\vec{\mathcal{E}}(\mathbf{r})$ and $\vec{\mathcal{B}}(\mathbf{r})$ denote the position-dependent complex electric and magnetic field amplitudes respectively[69,70]. The local enhancement of the optical chirality is calculated as $C(\mathbf{r})/|C_{CPL}|$, where $C_{CPL}$ represents the optical chirality of circularly polarized light without the plasmonic lattice.

## Data availability

Relevant data supporting the key findings of this study are available within the article and the Supplementary Information file. All raw data generated during the current study are available from the corresponding authors upon request.

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

## Acknowledgements

This work was supported by the National Key Research and Development Program of China (Grant No. 2021YFA1400700), the National Natural Science Foundation of China (Grants Nos. 62025507, 11934019, 92250301, 11721404, 62175254, 12174437, and 12204020), the Strategic Priority Research Program (Grant No. XDB28000000) of the Chinese Academy of Sciences, China Postdoctoral Science Foundation (Grant No. 2022M710234). We thank Duanlu Zhou, Hanjie Zhu, and Sibo Guo for their helpful discussions.

## Author contributions

X.X., C.W., K.J. and Q.G. supervised the research project. L.Y. designed the experiments and fabricated the plasmonic lattices. Y.H. and X.H. contributed materials and performed the transfer. L.Y., Y.Y., B.F., J.Y., D.D., S.S., S.Y., R.Z., H.L., Z.Z. and X.X. performed the optical measurements. L.Y. performed the simulations and calculations. L.Y., J.Y. and X.X. analyzed the data and wrote the manuscript with substantial contributions from all the others.

## Competing interests

The authors declare no competing interests.
