## [Peer Review File · Nature Communications]

Revealing broken valley symmetry of quantum emitters in WSe₂ with chiral nanocavitiesEditorial Note: Parts of this Peer Review File have been redacted as indicated to remove third-party material where no permission to publish could be obtained.

Reviewers' comments:

Reviewer #1 (Remarks to the Author):

In “Revealing broken valley symmetry of quantum emitters in WSe₂ with chiral nanocavities”, the authors present low-temperature photoluminescence measurements of quantum emitters in bare WSe₂ monolayers and WSe₂ monolayers coupled to chiral plasmonic cavities under different applied out-of-plane magnetic fields and helicity-resolved photon collection. The authors demonstrate that: i) the QEs in WSe₂ can couple to the chiral plasmon resonance of their nanoantennae with C₃ symmetry; ii) WSe₂ QEs do not retain the spin-valley locking characteristic of other excitonic transitions in monolayer WSe₂, as shown by the measured degree of circular polarisation in the photon emission of these localised excitons; iii) when the QEs are coupled the chiral plasmonic nanocavity, the helicity of the emitted photons is independent of the momentum (valley) of the carriers forming the exciton.

In general, I find the methods and experimental results enough to support some of the claims of the authors. However, although I agree that there might still be some debate in the community about the precise origin of QEs in WSe₂, in my opinion the manuscript does not contain the level of novelty, advance, and impact adequate for the broad audience of Nature Communications. Among the different theoretical models that aim to describe the origin of QEs in WSe₂, hybridisation of defect states with dark excitons (reference 18 in the manuscript) is, to the best of my knowledge, the only model that so far has been able to predict many of the physical properties of these QEs (such as their fine-structure splitting, emission lifetimes, exciton g-factors, and polarisation dependence). Indeed, as the authors point out in the manuscript, such a hybridisation between the defect states and the dark exciton has already been shown experimentally in Nature Communications 13, 7691 (2022). Therefore, I do not recommend the publication of the current article in Nature Communications. I believe the article is better suited for a more specialised journal.

Finally, below are my detailed comments:

1. With the helicity-resolved measurements under applied out-of-plane magnetic fields, the authors show that the helicity of photon emission for QEs coupled to the chiral plasmonic cavity is the same for the optical transitions involving the two valleys (in other words, that the polarisation state of the emitted photons is modulated by the chiral nanocavity instead of the valley-dependent optical selection rules). Based on these observations, the authors claim that the deterministic cavity-dependent circularly polarised single-photon emission of their coupled QEs (with a DCP of up to 60%) provides new opportunities for future quantum applications. I think the authors could expand this claim and clarify what type of applications they envision for such a DCP and the fact that, although they obtain single-photon emission with a certain degree of circular polarisation, the helicity of the emitted photons isn't linked to any quantum degree of freedom of the host material (i.e., spin or valley).

2. In line 36 of the current manuscript, the authors state: “Unlike the valley excitons, the hybridization with the defect state breaks the spin-valley locking, thus dramatically increases the transition oscillation strength, resulting in a very efficient single-photon emission.” Do the authors mean that the hybridisation leads to an increase of the oscillator strength of the single-photon emitters? If so, I think that the sentence could be written in a slightly different way, since currently it seems to suggest that the oscillator strength increases due to the breaking of the spin-valley locking.

3. In line 70, the authors state that the emission from delocalised neutral and charged excitons is quenched in their samples, which they attribute to a nonradiative energy transfer to the Au and conductive ITO substrates. Do the authors understand why there isn't quenching for the localised excitons? If so, adding a sentence explaining this could be beneficial for the reader.

4. The authors show that, in addition to the Zeeman splitting of the fine-structure emission doublets, the application of an out-of-plane magnetic field leads to a fast suppression of the emission from the high-energy peak in the emission doublet. The authors argue that such a suppression of the photon emission can't be exclusively attributed to the thermalisation of the excited carriers between the Zeeman-split peaks, and use as an example the fact that in In(Ga)As/(Al)GaAs quantum dots, both peaks do not exhibit attenuation even with a magnetic field up to 28 T. Although I agree that thermalisation may not be enough to explain the observed decrease in PL intensity for the high-energy peak, I think the comparison with III-V quantum dots isn't probably the best due to the different g-factors of these systems (generally a factor 3-4 smaller). Also, I think it would be great if the authors could add a 'rough' estimate of the PL quenching expected for pure thermalisation to justify their claim. Also, I note that other experimental reports do not show such a rapid decrease (or even disappearance) of the high-energy peak for QEs with similar g-factors (see for example Nature Nanotechnology 10, 503 (2015); npj 2D Materials and Applications 4, 2 (2020)).

5. In Figure 5b, the authors fit the experimental output power of a QE as a function of the applied magnetic field with the calculated output Φ_{out} as a function of the hybridisation strength, and show a good agreement with the experimental data. It isn't completely clear to me how the authors relate the magnetic field (top axis) to the hybridisation strength (bottom axis). I think that this should be discussed in the main manuscript at the relevant point in the discussion of the figure.

6. In line 165, the authors state: "It can be seen that the hybridization strength increases about orders of magnitude as the magnetic field linearly increases." I think there's a typo here and the authors forgot to include the order of magnitude of the increase (around 4?).

7. In line 202, the authors say: "For our system, the lifetime of deterministic circularly polarized single photon output could be switched from picosecond with QE radiation to femtosecond QE-CPR radiation." I think it would be good if the authors explicitly indicated that this a theoretical estimate based on their calculations.

Reviewer #2 (Remarks to the Author):

The paper titled "Revealing broken valley symmetry of quantum emitters in WSe₂ with chiral nanocavities" by Xiulai Xu et al. shows a detailed experimental investigation of how non-classical emitters in 2D WSe₂ monolayers break the valley symmetry by interacting with the chiral plasmonic nanocavities.

The manuscript is properly organized, presents novel results, and is written with attention to detail, especially given the extensive supplementary material. The paper is well-written and contains, to the best of my knowledge, a good list of appropriate references. The figures are clear and easy to understand.

The only aspect, which in this the paper appears to be somewhat lacking, is the clarity of the motivation, both from the fundamental and from the applied points of view.

1. The authors mention that it is surprising that the polarization state of emitted photons for neutral emitters is modulated by the chiral nanocavities instead of the valley-dependent optical selection rules. From the abstract and the introduction, it is not clear what is surprising about this behaviour, as it appears logical for neutral emitters. The authors mention that there has been a debate on the origin of excitons for these emitters referencing a review paper, although that review does not appear to focus on any controversy. Maybe the authors could expand the context of this question and clarify further why this is interesting.

2. The paper lacks any outlook. The authors very briefly mention that this system, provided the purity

is further improved, could become an “ideal platform for integrated single-photon-based plasmonic photonics”. It would be very useful for broad readership of Nature Communications, if the authors explained why they believe it, or otherwise dropped this claim. Including a proper outlook with a brief outline of follow-up research that potentially has either fundamental or applied value, would also significantly improve the submission.

Reviewer #3 (Remarks to the Author):

Yang et al. report an anomalous behavior in the emission polarization under linear and circular excitation of quantum emitters in WSe₂ coupled to a chiral plasmonic lattices under magnetic fields up to 9 Tesla. They explain their observations based on a recent theory where a hybridization between localized defect states and dark excitons was claimed to be the origin of quantum emitters in WSe₂. Furthermore, the chiral plasmonic lattice modifies the circular polarization of the emission which the authors find consistent with the theoretical prediction of Ref. 18.

I find the evidence provided and the corresponding arguments presented in support of their conclusion to be rather tenuous. They also seem to focus on a few emitters which show this behavior, and it is not clear why other emitters do not. As it is well-reported that there can be significant sample-to-sample variations and also emitter-to-emitter variations, such a claim needs to be backed by much more systematic study to rule out statistical fluctuations. For this reason, I cannot recommend publication of this manuscript.

Below I comment on two main evidence that the authors provide for their claim.

The first evidence that the authors provide for “intervalley defect exciton” is that the emission under magnetic field is primarily from the lowest energy peak split peak, which is a well-studied effect and often attributed to some sort of thermalization between the split peaks. The authors provide the example of InGaAs quantum dots to counter this argument of thermalization where both split peaks are observed even at 20 Tesla. However, comparing intensities across different material systems is tricky, for example, in carbon nanotubes a similar behavior as WSe₂ is observed i.e., the lowest energy peak dominates. This not only depends on details of phonons, etc involved but also the excitation conditions i.e., resonance versus non-resonant. This experiment was done far from resonance (532 nm laser for 775 nm emission) and could be repeated at quasi-resonance excitation of WSe₂ neutral exciton resonance.

The second observation that the authors make is that certain emitters show anomalous circular polarized emission under magnetic field and linear excitation. In particular, the helicity of emission is not very sensitive to magnetic field in the region where the chiral plasmonic lattice resonance is present. For example, as shown in figure S10a, at negative B-fields and under linear excitation, the emission is still slightly sigma+ when it is expected to be sigma- (as observed outside of the chiral plasmonic resonance). This apparent anomalous behavior of observing opposite circular polarization than expected in the presence of chiral plasmonic resonance was attributed to intervalley defect exciton. However, it is expected that the circular dichroism arising from their chiral plasmonic lattice will modify the polarization of emission (and excitation) and might selectively enhance one helicity in the far-field. I do not see the need to invoke “intervalley defect exciton” for this behavior. The behavior would be considered anomalous only if the free exciton coupled to a similar chiral plasmonic resonance did not show it.

The authors detected the DCP on a circular collection basis. How about a linear collection basis under zero magnetic field? Do the doublet structures have linear polarization only for QE 3, 4 but not for QE 5, 6, 7?

Response to Reviewers' Comment and Revisions

Reviewers' comments:

Reviewer #1 (Remarks to the Author):

In “Revealing broken valley symmetry of quantum emitters in WSe₂ with chiral nanocavities”, the authors present low-temperature photoluminescence measurements of quantum emitters in bare WSe₂ monolayers and WSe₂ monolayers coupled to chiral plasmonic cavities under different applied out-of-plane magnetic fields and helicity-resolved photon collection. The authors demonstrate that: i) the QEs in WSe₂ can couple to the chiral plasmon resonance of their nanoantennae with C₃ symmetry; ii) WSe₂ QEs do not retain the spin-valley locking characteristic of other excitonic transitions in monolayer WSe₂, as shown by the measured degree of circular polarisation in the photon emission of these localised excitons; iii) when the QEs are coupled the chiral plasmonic nanocavity, the helicity of the emitted photons is independent of the momentum (valley) of the carriers forming the exciton.

In general, I find the methods and experimental results enough to support some of the claims of the authors. However, although I agree that there might still be some debate in the community about the precise origin of QEs in WSe₂, in my opinion the manuscript does not contain the level of novelty, advance, and impact adequate for the broad audience of Nature Communications. Among the different theoretical models that aim to describe the origin of QEs in WSe₂, hybridisation of defect states with dark excitons (reference 18 in the manuscript) is, to the best of my knowledge, the only model that so far has been able to predict many of the physical properties of these QEs (such as their fine-structure splitting, emission lifetimes, exciton g-factors, and polarisation dependence). Indeed, as the authors point out in the manuscript, such a hybridisation between the defect states and the dark exciton has already been shown experimentally in Nature Communications 13, 7691 (2022). Therefore, I do not recommend the publication of the current article in Nature Communications. I believe the article is better suited for a more specialised journal.

Reply: We thank the Reviewer for the summary and comments on our results. As the referee agreed that there are debates on the real origin of QEs in WSe₂, which is very important to understand the recombination physics and to explore the potential applications. Furthermore, as referee mentioned the reference 18 paper is the only model so far which has explained many experimental work being published from different groups. Therefore it is highly desired to have a straightforward and solid experiment to prove the theoretical model and provide experimental evidence for origin of the QEs. To demonstrate the novelty, advance and impact of our work, we would like to emphasis the importance and innovation of our results from the following three points. At the same time, we have also made the corresponding modifications in the revised manuscript to make them more clearly described.

(1) First of all, as agreed by the reviewer, the precise origin of QE has always been controversial.

Whether QEs in WSe₂ inherit the physics of valley is very essential for understanding the physics and explore its applications, and is timely for such a large group of researchers working on this subject recently. If these properties are inherited, the defect state excitons can be regarded as valley excitons to achieve various valley related applications, such as valley and spin Hall effects. If these characteristics are not available, it is of great significance to explore the precise origin and intrinsic physical properties of these quantum emitters for their applications in quantum light sources and integrated photonics, due to the excellent optical properties and two-dimensional characteristics, such as flexible external control. In this work we experimentally demonstrated the broken valley symmetry of quantum emitters in WSe₂ with chiral nanocavities, particularly the anomalous chiral photon emission in a magnetic field with Faraday configuration, which has also been explained clearly with the model. **Therefore we believe this is the first paper to prove the model with single QEs as explained in the following point.**

Action taken: We have added more descriptions to show the significance of studying the physical properties of QEs in monolayer. (page 1, paragraph 1 and page 2, paragraph 1 with highlight in red)

(2) It is true that the results in Nature Communications 13, 7691 (2022) show the hybridization between the defect states and the dark exciton **with observing the PL intensity enhancement for different charging states**, which has been explained by the hybridization model. However, the valley polarization has not been investigated at all, the oscillation strength increases could be happened in most cases when bring two energies in resonance. Furthermore none single quantum emitters have been investigated. Therefore no solid conclusion has been made in the reference as they conclude in there Conclusion ‘the hybridized state we observe is **likely** key for the operation of single quantum emitters in WSe₂’. **However, in this work we designed the chiral nanocavities to enhance the emission from QEs, demonstrated unambiguously, the broken valley symmetry of single QEs, and clearly proved the theory as predicted in reference (Physical Review Letters 123, 146401 (2019)). Very interesting results are the same polarization in two PL branches under a magnetic field, which is counter intuitive. Furthermore we also explained clearly the PL disappearing of the high energy branch in magnetic field, which has not been discussed in detail before.** In the revised manuscript, in order to show the difference and importance of our experiments, we have stated that the existence of valley polarization in defect states **has not been experimentally confirmed until now, especially in the case of a single quantum emitter.** Furthermore, the potential applications of the intervalley defect excitons remain to be explored. We illustrate the importance of optical nanocavities in reading out the information of QE itself and in exploring new quantum optics devices. We highlight the importance of using chiral plasmonic nanocavity in solving those problems.

Action taken: We have added more descriptions to show the importance of our work in exploration of QE valley polarization and novel quantum optics devices by using nanocavities. (abstract; page 3, paragraph 1 with highlight in red)

(3) Last but not least, we developed a technique for control of light-matter interaction at the level of single quanta, indicating a potential application in chiral quantum optics. The

polarization state of the output single photon can be switched by the chirality of nanocavity while the electron spin of the single defect exciton state remains constant for the intervalley defect exciton. That is to say, we lock the spin angular momentum of the output photon of an exciton state with a definite spin by chiral nanocavities, which can be used to realize many new photonics devices, such as optical switch based on single chiral photon. It can also be embedded into nano-photonics devices such as waveguides or optical circulators based on chiral coupling to achieve directional output of spin states.

Action taken: We have added more discussions to elaborate the novelty of our work in applications of quantum optics. (abstract; page 3, paragraph 2 with highlight in red; summary)

Finally, below are my detailed comments:

1. With the helicity-resolved measurements under applied out-of-plane magnetic fields, the authors show that the helicity of photon emission for QEs coupled to the chiral plasmonic cavity is the same for the optical transitions involving the two valleys (in other words, that the polarisation state of the emitted photons is modulated by the chiral nanocavity instead of the valley-dependent optical selection rules). Based on these observations, the authors claim that the deterministic cavity-dependent circularly polarised single-photon emission of their coupled QEs (with a DCP of up to 60%) provides new opportunities for future quantum applications. I think the authors could expand this claim and clarify what type of applications they envision for such a DCP and the fact that, although they obtain single-photon emission with a certain degree of circular polarisation, the helicity of the emitted photons isn't linked to any quantum degree of freedom of the host material (i.e., spin or valley).

Reply: We thank the Reviewer for the important suggestions. The QEs with a DCP of up to 60% which independent the transition channels demonstrate the realization of cavity-dependent circularly polarized single-photon output. As discussed above, the polarization state of the output single photon can be switched by the chirality of nanocavity while the electron spin of the single defect exciton state remains constant for the intervalley defect exciton, which can be used to realize many new photonics devices, such as optical switch based on single chiral photon. It can also be embedded into nano-photonics devices such as waveguides or optical circulators based on chiral coupling to achieve directional output of spin states.

Action taken: We have added more discussions to elaborate the applications of our work for future quantum optics. (abstract; page 3, paragraph 2 with highlight in red; summary)

2. In line 36 of the current manuscript, the authors state: "Unlike the valley excitons, the hybridization with the defect state breaks the spin-valley locking, thus dramatically increases the transition oscillation strength, resulting in a very efficient single-photon emission." Do the authors mean that the hybridisation leads to an increase of the oscillator strength of the single-photon emitters? If so, I think that the sentence could be written in a slightly different way, since currently it seems to suggest that the oscillator strength increases due to the breaking of the spin-valley locking.

Reply: We thank the Reviewer for identifying this confusion and we changed the corresponding

description in the revised manuscript.

Action taken: We have changed the sentences in the revised manuscript to the following:” In contrast to valley excitons, the hybridization with the defect state not only breaks the spin-valley locking, but also dramatically increases the transition oscillation strength, resulting in an efficient photon emission”. (page 2, paragraph 2 with highlight in red)

3. In line 70, the authors state that the emission from delocalised neutral and charged excitons is quenched in their samples, which they attribute to a nonradiative energy transfer to the Au and conductive ITO substrates. Do the authors understand why there isn't quenching for the localised excitons? If so, adding a sentence explaining this could be beneficial for the reader.

Reply: We thank the Reviewer for pointing out this confusion. At low temperature, the excitons primarily populate in the lowest energy state which is dark state for WSe₂. This long-lived dark state is more likely to have nonradiative energy transfer with the Au or ITO substrate, which will enhance the energy relaxation rate of low-lying dark state. Due to the enhanced relaxation rate of dark state, the population of both dark and bright states will reduce due to the phonon scattering between the two states, causing the quenching of bright excitons emission (neutral and charged excitons). However, the defect states that are lower in energy than the dark states do not experience this quenching effect.

Action taken: We have changed the sentences in the revised manuscript to the following:” It is notable that the emissions from valley neutral excitons and trions are almost quenched in our sample. This is because the nonradiative energy transfer between the Au or ITO substrate and long-lived low energy dark states increases the relaxation rate of dark state in WSe₂, thus reducing the population of both dark and bright exciton states by phonon scattering between two states. However, the energy of defect states is lower than that of dark states of conduction band. As a result, the defect states do not experience this quenching effect”. (page 4, paragraph 2 with highlight in red)

4. The authors show that, in addition to the Zeeman splitting of the fine-structure emission doublets, the application of an out-of-plane magnetic field leads to a fast suppression of the emission from the high-energy peak in the emission doublet. The authors argue that such a suppression of the photon emission can't be exclusively attributed to the thermalisation of the excited carriers between the Zeeman-split peaks, and use as an example the fact that in In(Ga)As/(Al)GaAs quantum dots, both peaks do not exhibit attenuation even with a magnetic field up to 28 T. Although I agree that thermalisation may not be enough to explain the observed decrease in PL intensity for the high-energy peak, I think the comparison with III-V quantum dots isn't probably the best due to the different g-factors of these systems (generally a factor 3-4 smaller). Also, I think it would be great if the authors could add a 'rough' estimate of the PL quenching expected for pure thermalisation to justify their claim. Also, I note that other experimental reports do not show such a rapid decrease (or even disappearance) of the high-energy peak for QEs with similar g-factors (see for example Nature Nanotechnology 10, 503 (2015); npj 2D Materials and Applications 4, 2 (2020)).

Reply: We thank the Reviewer for the helpful suggestions that comparing intensities across

different material systems is not reasonable, similar to the Question 1 of Reviewer #3. We removed the comparison with In(Ga)As/(Al)GaAs quantum dots and we use the Fermi–Dirac statistics to show that thermalization is indeed not the cause of high energy branch quenching.

In thermal equilibrium, the probability distribution of electrons in different energy levels is determined by Fermi-Dirac distribution:

$$f = \frac{1}{e^{(E-E_f)/kT} + 1}$$

where k is Boltzmann constant, T is temperature, E is the energy of energy levels and E_f is the energy of Fermi level.

Figure R1 The relationship between the positions of defect levels of low and high energy peaks and Fermi levels.

The Fermi–Dirac statistics indicates that the energy level with high energy has a lower probability of electron distribution. Due to the contribution of valley magnetic moment and atomic orbital magnetic moment to the valence band, the defect level of the lower energy peak under the magnetic field is the one with high energy, as shown in Figure R1. This means that the PL quenching should be observed at low energy peak, which is contrary to the experimental results. This phenomenon can be well explained by the hybridization. With the increase of magnetic field, the energy difference between the defect level of high energy peak and the dark states of conduction band with the same spin becomes larger, leading to the rapid weakening of hybridization, which is the main reason for the PL quenching.

Such a rapid decrease of the high-energy peak for QEs is not observed in other reports (Nature Nanotechnology 10, 503 (2015); npj 2D Materials and Applications 4, 2 (2020)), which might be due to that the monolayer is deposited on a smooth Si/SiO₂ substrate without any strain engineering in their experiments. In this case, the energy difference between the defect levels and the dark conduction band are large. The hybridization is not very sensitive to the shifts of energy levels or bands under magnetic field, thus there is no rapid disappearance or increase of one of them. In our experiments, strain is more likely to occur at the edge of the metal structure, which promotes the downshifts of conduction band and its proximity to the defect levels.

Action taken: We have removed the comparison with In(Ga)As/(Al)GaAs quantum dots and we have used the Fermi–Dirac statistics to show that thermalization is indeed not the cause of energy quenching in the revised manuscript. (page 4, paragraph 3 with highlight in red)

5. In Figure 5b, the authors fit the experimental output power of a QE as a function of the applied magnetic field with the calculated output Φ_{out} as a function of the hybridisation strength, and show a good agreement with the experimental data. It isn't completely clear to me how the authors relate the magnetic field (top axis) to the hybridisation strength (bottom axis). I think that this should be discussed in the main manuscript at the relevant point in the discussion of the figure.

Reply: We are sorry that we did not describe it more clearly. With the increase of the magnetic field, the energy difference between the dark states of conduction band and the defect level for the low energy branch of QE will decrease accordingly. This energy proximity directly leads to the increase of the hybrid strength, thus leading to the increase of the output power of QE. The energy difference varies linearly with the magnetic field, while the hybrid strength increases nonlinearly with the energy difference. Therefore, the magnetic field (top axis) in the figure actually represents the energy difference between the dark conduction band and the energy level. We use Figure 5b to explore the scale of this nonlinear hybridization rate with respect to the energy difference. And the energy difference is replaced by a magnetic field that is linear with it.

Action taken: We have added more descriptions and discussions about Figure 5b in revised manuscript. (page 7, paragraph 3 with highlight in red)

6. In line 165, the authors state: "It can be seen that the hybridization strength increases about orders of magnitude as the magnetic field linearly increases." I think there's a typo here and the authors forgot to include the order of magnitude of the increase (around 4?).

Reply: We thank the Reviewer for the helpful suggestion. As discussed in question 5, we have made a more appropriate description according to the reviewer's requirements in the revised manuscript.

Action taken: We have changed the sentences in the revised manuscript to the following: "We find that the experimental results fit well with the calculated results when the hybridization strength increases by about 4 orders of magnitude as the energy approaches". (page 7, paragraph 3 with highlight in red)

7. In line 202, the authors say: "For our system, the lifetime of deterministic circularly polarized single photon output could be switched from picosecond with QE radiation to femtosecond QE-CPR radiation." I think it would be good if the authors explicitly indicated that this a theoretical estimate based on their calculations.

Reply: We thank the Reviewer for the helpful suggestion. We converted the axis in the insert in Figure 5d into units of femtosecond in the revised manuscript. The population on femtosecond timescale shows a rapid oscillation decay within about 60 fs whenever in weak coupling or strong coupling condition, indicating an ultrafast energy transfer between the QE and plasmon field. It also indicates that the chiral plasmon field switches the polarization of the output photon in a very short time.

Action taken: We have changed the axis in the inset in Figure 5d into units of femtosecond in

the revised manuscript.

Reviewer #2 (Remarks to the Author):

The paper titled “Revealing broken valley symmetry of quantum emitters in WSe₂ with chiral nanocavities” by Xiulai Xu et al. shows a detailed experimental investigation of how non-classical emitters in 2D WSe₂ monolayers break the valley symmetry by interacting with the chiral plasmonic nanocavities.

The manuscript is properly organized, presents novel results, and is written with attention to detail, especially given the extensive supplementary material. The paper is well-written and contains, to the best of my knowledge, a good list of appropriate references. The figures are clear and easy to understand.

The only aspect, which in this the paper appears to be somewhat lacking, is the clarity of the motivation, both from the fundamental and from the applied points of view.

Reply: We really appreciate the Reviewer for the recognition of our manuscript, especially the novelty of our results. At the same time, we also appreciate the Reviewer's constructive comments, including research motivation, the fundamental importance and potential applications of our work. In order to make these more clearly, we have made the following changes in the revised manuscript:

(1) We have added the research motivation and the statement of the significance of studying the inherent physical properties of QE. Whether the QE inherits the physics of valley is very essential for its applications. If these properties are inherited, the defect state excitons can be regarded as valley exciton to achieve various valley related applications. If these characteristics are not available, it is of great significance to explore the precise origin and intrinsic physical properties of these quantum emitters for their applications in quantum light sources and integrated photonics, due to the excellent optical properties and two-dimensional characteristics.

Action taken: We have added more descriptions to show the significance of studying the physical properties of QEs in monolayer. (page 1, paragraph 1 and page 2, paragraph 1 with highlight in red)

(2) In order to emphasize the importance and difference of our experiments, we have stated in the revised manuscript that the existence of valley polarization in defect states has not been experimentally confirmed until now, especially in the case of a single quantum emitter. Furthermore, the potential applications of the intervalley defect excitons remain to be explored. We illustrate the importance of optical nanocavities in reading out the information of QE itself and in exploring new quantum optics devices. We highlight the importance of using chiral plasmonic nanocavity in solving those problems.

Action taken: We have added more descriptions to show the importance of our work in exploration of QE valley polarization and novel quantum optics devices by using chiral

plasmonic nanocavities. (abstract; page 3, paragraph 1 with highlight in red)

(3) We have added more descriptions about the future applications of our work and added the corresponding supporting literatures. We develop a technique for control of light-matter interaction at the level of single quanta, indicating applications in chiral quantum optics. The polarization state of the output single photon can be switched by the chirality of nanocavity while the electron spin of the single defect exciton state remains constant for the intervalley defect exciton. That is to say, we lock the spin angular momentum of the output photon of an exciton state with definite spin by chiral nanocavities, which can be used to realize many new photonics devices, such as optical switch based on single chiral photon. It can also be embedded into nano-photonics devices such as waveguides or optical circulators based on chiral coupling to achieve directional output of spin states.

Action taken: We have added more discussions to elaborate the novelty of our work in applications of quantum optics. (abstract; page 3, paragraph 2 with highlight in red; summary)

1. The authors mention that it is surprising that the polarization state of emitted photons for neutral emitters is modulated by the chiral nanocavities instead of the valley-dependent optical selection rules. From the abstract and the introduction, it is not clear what is surprising about this behaviour, as it appears logical for neutral emitters. The authors mention that there has been a debate on the origin of excitons for these emitters referencing a review paper, although that review does not appear to focus on any controversy. Maybe the authors could expand the context of this question and clarify further why this is interesting.

Reply: We thank the Reviewer for pointing out this confusion. We are sorry that we did not describe it more clearly. Whether the transition of a QE is dominated by the valley-dependent optical selection rules will lead to completely different experimental results. As discussed in Question 2 of Reviewer #3, the incomplete circular polarized emission can be caused by different mechanisms. If it is due to the valley polarization, then the decrease in DCP is caused by various relaxation processes, such as phonon-assisted intervalley scattering, where the absorption process is not enhanced by CPR with cross-polarized. The presence of CPR may strengthen or weaken the relaxation process, leading to a further reduction or increase in the DCP, but not to a reversal of the DCP. For example, the scattering of valley excitons can be reduced by the microcavity (Nature Photonics 11, 431 (2017) and Nature Photonics 11, 497 (2017)), but the valley polarization of a certain valley cannot be reversed. If it is not related to valley polarization, but the transition itself is not completely polarized. That means that the QE itself doesn't obey the selection rules strictly as the exciton angular momentum of QE is not a good quantum number. In this case, the transition of QE contains the components co-polarized and cross-polarized with CPR, then CPR can enhance not only the radiation, but also the co-polarized absorption process, resulting in a reversal of DCP. Our experiments show a surprising reversal of polarization of the QEs coupled to CPR comparing to the QE on substrate, that is, the nanocavity-dependent circularly polarized photon output, confirming the unique property of QE that is different from valley excitons and proving the absence of valley polarization.

We also thank the Reviewer for pointing out that the debate in the review paper refers to the origin of quantum confinement for QE rather than its physical properties. We have cited

appropriate literatures to clarify our topic in the related paragraph in the revised manuscript (page 1, paragraph 2 and page 2, paragraph 1).

Action taken: We have added more discussions to illustrate the fundamental difference between the cavity-dependent and the valley-dependent circularly polarized emission in the revised manuscript. We also added the detailed discussions in the Supplementary Information. (page 5, paragraph 2 and page 7, paragraph 1 with highlight in red; Supplementary Information, Figure S2)

2. The paper lacks any outlook. The authors very briefly mention that this system, provided the purity is further improved, could become an “ideal platform for integrated single-photon-based plasmonic photonics”. It would be very useful for broad readership of Nature Communications, if the authors explained why they believe it, or otherwise dropped this claim. Including a proper outlook with a brief outline of follow-up research that potentially has either fundamental or applied value, would also significantly improve the submission.

Reply: We thank the Reviewer for the helpful suggestion. We have added more descriptions about the future applications of our work and added the corresponding supporting literatures. As also discussed above, we develop a technique for control of light-matter interaction at the level of single quanta, indicating applications in chiral quantum optics. We lock the spin angular momentum of the output photon of an exciton state with definite spin by chiral nanocavities, which can be used to realize many new photonics devices, such as optical switch based on single chiral photon. It can also be embedded into nano-photonics devices such as waveguides or optical circulators based on chiral coupling to achieve directional output of spin states.

Action taken: We have added more discussions and outlook to elaborate the novelty of our work in applications of quantum optics and added specific examples to demonstrate the applications in integrated single-photon-based plasmonic photonics in the revised manuscript. (abstract; page 3, paragraph 2 with highlight in red; summary)

Reviewer #3 (Remarks to the Author):

Yang et al. report an anomalous behavior in the emission polarization under linear and circular excitation of quantum emitters in WSe₂ coupled to a chiral plasmonic lattices under magnetic fields up to 9 Tesla. They explain their observations based on a recent theory where a hybridization between localized defect states and dark excitons was claimed to be the origin of quantum emitters in WSe₂. Furthermore, the chiral plasmonic lattice modifies the circular polarization of the emission which the authors find consistent with the theoretical prediction of Ref. 18.

I find the evidence provided and the corresponding arguments presented in support of their conclusion to be rather tenuous. They also seem to focus on a few emitters which show this behavior, and it is not clear why other emitters do not. As it is well-reported that there can be significant sample-to-sample variations and also emitter-to-emitter variations, such a claim needs to be backed by much more systematic study to rule out statistical fluctuations. For this reason, I cannot recommend publication of this manuscript.

Reply: We thank the Reviewer for the comments. We are sorry that we did not explain clearly the universality of our experiments and the reliability of our conclusions. As stated by the Reviewer, only part of the QEs shows anomalous behavior. However, this exactly shows that our results are reliable, because not all of the QEs is coupled to the chiral plasmon field. As shown in the Figure 1e in the main text, the chiral field almost only exists around the edge of the chiral nanostructure, and the area distributed with the chiral field is much smaller than that of the entire unit cell, that is, it only accounts for a small part of the area of monolayer covered on the lattice. Although QEs are more likely to be generated at the edge due to the existence of strain, not all QEs can be located exactly in the local field with high optical chirality, and even some areas have strong plasmon field, but their optical chirality is very low, as shown in Figure 1d in the main text. Therefore, in our experiments, only part of the QEs can be coupled to the chiral plasmon field and show anomalous behavior, while the rest of the QE radiation behaves normally.

We admit that the Reviewer's consideration about sample-to-sample variations and emitter-to-emitter variations is reasonable. We will verify the reliability of our results from the following three aspects:

(1) The QEs we studied are all neutral defect excitons in WSe₂ monolayer, which can be confirmed by the fine structure splitting and similar g factors, as shown in Figure 3 in the main text and Figure S6 in Supplementary Information. So we confirm that the QEs with anomalous behavior are of the same type as the other QEs. We have described the species of excitons of QEs in the manuscript (page 4, paragraph 3 with highlight in blue).

(2) We have not only studied the properties of single quantum emitter, but we have also studied the collective behavior of a large number of QEs. As shown in Figure 4d and 4e in the main text, we measured the polarization-resolved magneto-photoluminescence spectroscopy of defect emission with broad peak under high power excitation. Compared with the uncoupled defect states on the substrate, the defect states on the nanostructures with different parameters show different degrees of abnormal behavior. The location of these anomalous peaks is related to the energy of CPR and the uneven distribution of defects. This anomalous radiation behavior with a large amount of QEs proves that our observations are not accidental.

(3) As suggested by the Reviewer, we did the corresponding statistics on the QE emissions. As shown in Figure R2, the spectra show the polarization-resolved photoluminescence from QEs at L1, L2 and substrate, respectively. To count the number of coupled and uncoupled QEs, we show only the results of 9T and -9T. Since only low energy peaks occur at high magnetic fields, each narrow linewidth single peak represents the radiation of a QE. As shown in Figure R2a, we observed a large number of narrow linewidth emissions at L1 due to the effective matching of defects energy and CPR mode. We performed statistics on the coupled QEs with anomalous behavior, that is, QEs with the same helicities under opposite magnetic fields, as shown by the blue arrows in Figure R2. We find that among 29 QEs, 14 QEs have anomalous behavior, indicating a high proportion of QEs coupled to CPR. This high proportion is highly correlated with the fact that QEs are more likely to be generated at the edge and that the chiral field is

mainly distributed at the edge of the nanostructure. QEs coupled to CPR can also be found at L2 as shown in Figure R2b, though the number of individual QEs with narrow linewidths is much lower than that at L1 due to poor resonance with the CPR mode. But at both L1 and L2, the results on the chiral nanostructure are in stark contrast to those on the substrate, as shown in Figure R2c. Our statistical results show that the proportion of QEs with anomalous behavior is quite high, and is not caused by statistical fluctuations.

Figure R2 Statistics on the QEs coupled to CPR. The spectra show the polarization-resolved

photoluminescence from QEs (a) at L1, (b) at L2 and (c) on ITO substrate, respectively. The results were measured under -9 T and 9 T magnetic field and were detected in the circular basis (black lines: σ^+ configuration; red lines: σ^- configuration). The black and blue arrows represent the QEs that are not coupled to CPR and QEs that are coupled to CPR, respectively.

Action taken: We have added a corresponding statement to illustrate that the anomalous radiation behavior with a large amount of QEs in Figure 4d in the main text proves that our observations are not coincidental. (page 7, paragraph 1 with highlight in red)

We have added the statistical result of QEs with anomalous behavior and added the corresponding statement to illustrate that this proportion is related to the distribution of chiral plasmon field in the revised manuscript. (page 6, paragraph 1 with highlight in red; Supplementary Information, Figure S13)

Below I comment on two main evidence that the authors provide for their claim.

1. The first evidence that the authors provide for “intervalley defect exciton” is that the emission under magnetic field is primarily from the lowest energy peak split peak, which is a well-studied effect and often attributed to some sort of thermalization between the split peaks. The authors provide the example of InGaAs quantum dots to counter this argument of thermalization where both split peaks are observed even at 20 Tesla. However, comparing intensities across different material systems is tricky, for example, in carbon nanotubes a similar behavior as WSe2 is observed i.e., the lowest energy peak dominates. This not only depends on details of phonons, etc involved but also the excitation conditions i.e., resonance versus non-resonant. This experiment was done far from resonance (532 nm laser for 775 nm emission) and could be repeated at quasi-resonance excitation of WSe2 neutral exciton resonance.

Reply: We thank the Reviewer for the helpful suggestions that comparing intensities across different material systems is not reasonable, just as the Question 4 of Reviewer #1 suggests. We removed the comparison with In(Ga)As/(Al)GaAs quantum dots and we use the Fermi–Dirac statistics to show that thermalization is indeed not the cause of stronger low energy peak emission, as discussed above in Question 4 of Reviewer #1. This is the result of the particularity of transitions from energy levels to different valleys and hybridization with dark bands for QEs in WSe2 monolayer.

Figure R3 The quasi-resonance excitation of WSe2 neutral defect exciton in the reference:

Nature Nanotechnology 14, 426-431 (2019). The excitation wavelength of laser is 752 nm (1.649 eV).

We are sorry that we haven't done the corresponding quasi-resonance excitation experiments, but we note that there have been experimental results on quasi-resonant excitation of WSe2 neutral defect exciton in the reference (Nature Nanotechnology 14, 426-431 (2019)). As shown in Figure R3, even under quasi-resonance excitation, the high energy peaks are significantly weaker than low energy peaks, and decay rapidly with the increase of the magnetic field. This further proves that thermalization is not the reason for the stronger intensity of lower energy peaks. Under quasi-resonance excitation, the intensity of high energy peak decreases rapidly with the magnetic field, which also indicates the existence of hybridization.

Action taken: We have removed the comparison with In(Ga)As/(Al)GaAs quantum dots and we have used the Fermi–Dirac statistics to show that thermalization is indeed not the cause of energy quenching in the revised manuscript. (page 4, paragraph 3 with highlight in red)

2. The second observation that the authors make is that certain emitters show anomalous circular polarized emission under magnetic field and linear excitation. In particular, the helicity of emission is not very sensitive to magnetic field in the region where the chiral plasmonic lattice resonance is present. For example, as shown in figure S10a, at negative B-fields and under linear excitation, the emission is still slightly sigma+ when it is expected to be sigma- (as observed outside of the chiral plasmonic resonance). This apparent anomalous behavior of observing opposite circular polarization than expected in the presence of chiral plasmonic resonance was attributed to intervalley defect exciton. However, it is expected that the circular dichroism arising from their chiral plasmonic lattice will modify the polarization of emission (and excitation) and might selectively enhance one helicity in the far-field. I do not see the need to invoke “intervalley defect exciton” for this behavior. The behavior would be considered anomalous only if the free exciton coupled to a similar chiral plasmonic resonance did not show it.

Reply: We thank the Reviewer for pointing out this confusion. The precise model of the QE in WSe2 will be very important for experimental interpretation and guidance of potential applications. As predicted in the previous literature (Physical Review Letters 121, 167402 (2018)), the intravalley defect states will have the same optical selection rules at K and K' points as the valley excitons. The precise physical model of QE and whether the transition of a QE is dominated by the valley-dependent optical selection rules will lead to completely different experimental results.

Figure R4 Polarization of output photon of QE coupled with CPR under different transition mechanisms. (a) The circularly polarized emission of transitions of QE in a circular basis

measurement, showing incomplete circularly polarized photon in each channel. (b) and (c) The transition processes with valley polarization dependent and independent respectively.

As shown in Figure R4, the incomplete circular polarized emission can be caused by different mechanisms. If it is due to the valley polarization, for example, for the intravalley defect excitons, then the decrease in DCP is caused by various relaxation processes, such as phonon-assisted intervalley scattering, where the absorption process is not enhanced by CPR with cross-polarized, as shown in Figure R4b. The presence of CPR may strengthen or weaken the relaxation process, leading to a further reduction or increase in the DCP, but not to a reversal of the DCP. For example, the scattering of valley excitons can be reduced by the microcavity (Nature Photonics 11, 431 (2017) and Nature Photonics 11, 497 (2017)), but the valley polarization of a certain valley cannot be reversed. If it is not related to valley polarization, but the transition itself is not completely polarized. That means that the QE itself doesn't obey the selection rules strictly as the exciton angular momentum of QE is not a good quantum number. In this case, the transition of QE contains the components co-polarized and cross-polarized with CPR, then CPR can enhance not only the radiation, but also the co-polarized absorption process as shown in Figure R4c, resulting in a reversal of DCP. Our experiments show a surprising reversal of polarization of the QEs coupled to CPR comparing to the QE on substrate, that is, the nanocavity-dependent circularly polarized photon output, confirming the unique property of QE that is different from valley excitons and intravalley defect excitons. Our work strongly confirms the model of intervalley defect exciton and the absence of valley polarization for QE in WSe₂.

Our work not only has important implications for the study of the properties of QE in 2D WSe₂ monolayer, but also develops a technique for applying intervalley defect excitons to chiral quantum optics at the level of single quanta. The polarization state of the output single photon can be switched by the chirality of nanocavity while the electron spin of the single defect exciton state remains constant for the intervalley defect exciton. That is to say, we lock the spin angular momentum of the output photon of an exciton state with a definite spin by chiral nanocavities, which can be used to realize many new photonics devices, such as optical switch based on single chiral photon. It can also be embedded into nano-photonics devices such as waveguides or optical circulators based on chiral coupling to achieve directional output of spin states. These intriguing applications all build on our understanding of intervalley defect excitons for QE in WSe₂.

Action taken: We have added more discussions to illustrate the fundamental difference between the intervalley defect exciton and the valley exciton in the revised manuscript. We also added the detailed discussions in the Supplementary Information. (page 5, paragraph 2 and page 7, paragraph 1 with highlight in red; Supplementary Information, Figure S2). We have added more discussions to elaborate the novelty of our work in applications of quantum optics based on the intervalley defect exciton. (abstract; page 3, paragraph 2 with highlight in red; summary)

3. The authors detected the DCP on a circular collection basis. How about a linear collection basis under zero magnetic field? Do the doublet structures have linear polarization only for QE 3, 4 but not for QE 5, 6, 7?

Reply: We thank the Reviewer for the helpful suggestion. We have measured the PL-intensity plot of QEs with detected in linear basis under zero magnetic field as shown in Figure R5. A pair of cross-linearly polarized spectral doublet is observed for the uncoupled QE3 due to the electron-hole exchange interaction as shown in the top plot in Figure R5, consistent with those reported in the literatures. For the QE coupled to CPR, we do not observe this cross-linearly polarized spectral doublet similar to QE3. Instead, we observe a pair of elliptically polarized spectral doublet with the same polarization properties, such as the measurement results in QE6, as shown in the lower plot in Figure R5. This indicates that both emission modes of fine-structure split doublet at zero magnetic field are coupled to chiral plasmon, and that their polarization properties are strongly modulated by the chiral field. For the elliptically polarized light emitted by the coupled QE, the degree of circular polarization of the output photon can be further improved by the chiral plasmon field with higher optical chirality.

Figure R5 Contour plot of linear polarization-dependent PL spectra of QE3 (top) and QE6 (bottom) under zero magnetic field.

Action taken: We have added the experimental results measured in linear basis under zero magnetic field to the Supplementary Information and added more discussions about the difference between the coupled and uncoupled QE (Supplementary Information, Figure S16).

REVIEWER COMMENTS

Reviewer #1 (Remarks to the Author):

I have read through the authors' rebuttals to all the reviewers' comments. In general, I think that the authors have made a good effort to reply to all the concerns and doubts raised by the reviewers. However, my main criticism remains the same as in the previous version. In my opinion, the manuscript does not contain the level of novelty, advance, and impact adequate for the broad readership of Nature Communications. I think that the results of the authors would fit better in a more specialised journal.

Some additional comments:

- 1) I'm still not fully convinced about the nano-photonics devices that the authors envision as a potential application of their findings. As the authors state in their reply to my previous comment #1, "the polarisation state of the output single photon can be switched by the chirality of the nanocavity while the electron spin of the single defect state remains constant for the intervalley defect exciton". It isn't completely clear to me how this can be used as an "optical switch based on single chiral photon". For a defined cavity-emitter configuration, the polarisation of the output photon is set. How do the authors envision using this as a switch? By reversing the orientation of the applied vertical magnetic field? If so, how viable would doing that be in a nanophotonic device?
- 2) I agree with the authors that embedding these hybrid emitter/cavity systems in waveguides could result in directional coupling of the emitted photons. However, if I understand correctly, these directional coupling would be the same for the two different states, as the emitters arising from the two different valleys would couple to the same direction regardless of the electron spin. Therefore, I'm not convinced that the proposed system can be used to "achieve directional output of spin states."

Reviewer #2 (Remarks to the Author):

In my opinion, the manuscript has now been improved significantly and appears to meet most of the requirements for Nature Communications. I however would like to refrain from strongly endorsing the submission, as the impact of the work still appears somewhat unclear.

Reviewer #3 (Remarks to the Author):

The authors have addressed some of my comments to a certain degree of satisfaction but there are still many gaps in showing that the purported model of hybridization is at play here. Moreover, Nature Communications 13, 7691 (2022) has provided experimental evidence for the model that the authors are trying to confirm. As a result, I still maintain that this work is suitable for a more specialized journal.

The authors should at least try to compare and contrast the effect of their chiral photonics cavity has on quantum emitters versus free excitons to further support their reasoning.

Response to Reviewers' Comment and Revisions

Reviewers' comments:

Reviewer #1 (Remarks to the Author):

I have read through the authors' rebuttals to all the reviewers' comments. In general, I think that the authors have made a good effort to reply to all the concerns and doubts raised by the reviewers. However, my main criticism remains the same as in the previous version. In my opinion, the manuscript does not contain the level of novelty, advance, and impact adequate for the broad readership of Nature Communications. I think that the results of the authors would fit better in a more specialised journal.

Reply: We thank the Reviewer for the appreciation of our reply. We are sorry that we have not convinced the Reviewer for the importance and the novelty of our work. However, we would like to emphasize again on the novelty and impact particularly with our new experiments with MoSe₂ layer.

Here in this work, we experimentally demonstrate **for the first time the breaking of valley symmetry in a single quantum emitter** in our manuscript. The valley physics of 2D layers have attracted wide attention of researchers. There has been a lot of work to try to verify those properties in those quantum emitters, as discussed in paragraph 2 in the introduction. Our work gives a conclusion that valley physics is not inherited in the **single quantum emitters** in WSe₂, which is most commonly studied in TMDs materials. Our work clears up this confusion and provides solid and unambiguous insights on the microscopic origin of quantum emitters in 2D semiconductors, which will dramatically change the understanding of this new type of quantum light source.

To further confirm our main claims on the breaking of valley symmetry in WSe₂ quantum emitters, we **fabricated new devices covered by MoSe₂ monolayer and made the corresponding measurements**, as shown in Figure R3 and discussed in Question 1 of Reviewer #3. The experiments of the coupling between MoSe₂ monolayer and chiral plasmonic lattices show that the valley-dependent selection rules are preserved in intravalley defect excitons. This further strengthens the validity of intervalley defect exciton model and our conclusions.

Action taken: Addition experimental results of MoSe₂ monolayer have been added in Supplementary Information (Supplementary Fig. 18) and discussed in the revised manuscript (page 6, paragraph 3 with highlighted in red).

Some additional comments:

1. I'm still not fully convinced about the nano-photonics devices that the authors envision as a potential application of their findings. As the authors state in their reply to my previous comment #1, "the polarisation state of the output single photon can be switched by the chirality of the nanocavity while the electron spin of the single defect state remains constant for the intervalley defect exciton". It isn't completely clear to me how this can be used as an "optical

switch based on single chiral photon". For a defined cavity-emitter configuration, the polarisation of the output photon is set. How do the authors envision using this as a switch? By reversing the orientation of the applied vertical magnetic field? If so, how viable would doing that be in a nanophotonic device?

Reply: We thank the Reviewer for pointing out this confusion and we are glad to describe it more clearly. The optical switch here we refer to is based on the strongly coupled system. For our system, the plasmonic nanocavity has an ultrasmall mode volume and can also strongly interact with a single quantum emitter. In principle, optimization of chiral nanocavity such as using a single plasmonic nanocavity and controlling the position of the quantum emitter can achieve a strong coupling with the system. In prospect of future, it is possible to achieve chiral single photon switching under strong coupling regime, similar to the demonstration discussed in Nature Photonics, 6, 605–609 (2012). We are sorry we described the prospective with such a confusion.

Action taken: We have changed the sentences in the revised manuscript to the following to express the potential applications more precisely: ".....the system can be used to demonstrate many new photonics devices, such as single photon sources with deterministic chirality, or chiral single-photon switches when the system is optimized in a strongly coupled regime" (page 9, paragraph 4 with highlighted in red).

The descriptions in abstract about the applications have also been revised correspondingly.

2. I agree with the authors that embedding these hybrid emitter/cavity systems in waveguides could result in directional coupling of the emitted photons. However, if I understand correctly, these directional coupling would be the same for the two different states, as the emitters arising from the two different valleys would couple to the same direction regardless of the electron spin. Therefore, I'm not convinced that the proposed system can be used to "achieve directional output of spin states."

Reply: We thank the Reviewer for pointing out this confusion and we are sorry that we did not describe it more clearly as well. It is true that, the emitters arising from the two different valleys would couple to the same direction regardless of the electron spin when embedding the coupled system in waveguides. However, here we assume the system works at high magnetic fields. In this case, only the low energy peaks can be observed in the PL spectra due to the rapid weakening of hybridization for the high energy peaks as we have discussed in the manuscript. Thus, we only need to focus on the radiation from the low energy branch. For example, in a positive magnetic field, only the transition of the defect level and valence band with spin up can be experimentally observed as shown in Figure R1(a). When the intervalley defect exciton is in a σ^+ polarized plasmon hot spot, the output of the spin state with spin up will propagate to the left, as shown in Figure R1(b), and to the right when it is in σ^- polarized plasmon hot spot as shown in Figure R1(c). Therefore, the proposed system can be used to directionally output the photons from a certain spin state by using the plasmonic nanocavity with different chirality, which can be used in future photonic devices.

Figure R1 Diagram of the directional output of the spin state of coupled intervalley defect exciton/chiral plasmonic nanocavity system embedded into waveguide based on chiral coupling. (a) Bare intervalley defect excitons. (b) Intervalley defect excitons in σ^+ polarized plasmon hot spot. (c) Intervalley defect excitons in σ^- polarized plasmon hot spot.

Action taken: We have changed the sentences in the revised manuscript to the following: “Directional output of one certain spin state at a high magnetic field can be achieved when the coupled system is embedded into nano-photonics devices such as waveguides or optical circulators based on the chiral coupling.” (page 9, paragraph 4 with highlighted in red).

The descriptions in abstract about the applications have also been revised correspondingly.

Reviewer #2 (Remarks to the Author):

In my opinion, the manuscript has now been improved significantly and appears to meet most of the requirements for Nature Communications. I however would like to refrain from strongly endorsing the submission, as the impact of the work still appears somewhat unclear.

Reply: We are grateful for the positive comments from the Reviewer.

Reviewer #3 (Remarks to the Author):

1. The authors have addressed some of my comments to a certain degree of satisfaction but there are still many gaps in showing that the purported model of hybridization is at play here.

Reply: We thank the Reviewer for the comments. In our manuscript, the model of hybridization

i.e. the intervalley defect exciton model for the quantum emitters in WSe₂ is crucial for explaining the anomalous circularly polarized emission. To further demonstrate the necessity of the hybridization model and the uniqueness of the intervalley defect exciton, we **have fabricated and measured new devices covered with MoSe₂ monolayer which has intravalley defect excitons**. Different from intervalley defect excitons in WSe₂, intravalley defect excitons in MoSe₂ monolayer are unrelated to hybridization. Below are the detailed discussion and results.

[redacted]

Editorial Note: Figure R2 has been redacted to remove third-party material where no permission to publish could be obtained.

As predicted in the previous literature (Physical Review Letters 121, 167402 (2018)), for the MoSe₂ monolayer, “transitions between the valence band maximum and the unoccupied defect bands have the same optical selection rules as the bulk valence band maximum-to-conduction band minimum transition in the vicinity of the K and K’ points”, as shown in Figure R2. This is a reasonable calculation based on the intravalley transition for MoSe₂ monolayer, because the valley exciton in MoSe₂ is “bright” since optical intravalley transitions between valence band maximum and conduction band minimum are spin allowed. Therefore, unlike WSe₂, the defect states in MoSe₂ do not need to undergo cross-valley hybridization to produce a strong photons emission. Thus, the transition process of the defect exciton in MoSe₂ is valley polarization dependent. The presence of chiral plasmon resonance may strengthen or weaken the relaxation process, leading to a further reduction or increase in the degree of circular polarization, but not to a reversal of the degree of circular polarization, as discussed in Supplementary Fig. 2.

Figure R3 (a) shows optical microscope images of MoSe₂ monolayer covered on the plasmonic lattices L1-L4 and the bottom is the corresponding PL mapping. Figure R3 (b) shows PL spectra of the hybrid structures, we can find that the emission peaks of excitons are modified by the chiral plasmon resonances compared to the measured reflection spectra as shown in Figure 2b in the main text due to the resonance with the chiral plasmon mode.

Figure R3 Optical characterizations of the interaction between the chiral plasmon and the MoSe₂ monolayer. (a) Optical microscope images of MoSe₂ monolayer covered on the plasmonic lattices L1-L4 and the corresponding PL mapping. (b) The room temperature PL spectra of the hybrid structures. The measurements in (a) and (b) were taken at room temperature. (c) and (d) Measured magnetic field dependent degree of circular polarization (DCP) of the MoSe₂ monolayer covered on plasmonic lattices L1-L4 and substrates, respectively. The corresponding measurements were performed with an excitation by a linearly polarized light at 4.2 K. The gray dashed lines in (c) mark the points where degree of circular polarization equals to zero. (e) The measured circular dichroism (CD) spectra of plasmonic lattices L1-L4.

Figure R3 (c) and (d) show the measured the magnetic field dependent degree of circular polarization of the MoSe₂ monolayer covered on plasmonic lattices L1-L4 and substrates, respectively, where the emissions with energy below 1.63 eV come from the radiation of massive defects under high power excitation. We find that, in contrast to the results on the Au film and ITO substrate, the circularly polarized emission of the monolayer on the plasmonic lattices is modified only at low magnetic fields. We trace the points where DCP equals to zero as the gray dashed lines in Figure R3(c) show and compared them with the CD spectra in Figure R3(e). We find that the anomalous emissions at low magnetic fields are caused by the radiation of the whole chiral plasmon resonance in the far field, which can be attributed to the strong electron-hole exchange interaction and weak thermal relaxation of high energy peaks. As the magnetic field increases, the electron-hole exchange interaction in the defect excitons is

overcome and the circularly polarized emissions are recovered. Meanwhile, with the increase of splitting, the intensity of high energy peaks gradually weakens due to the strengthening of the thermal relaxation. Due to the valley-dependent selection rules, the circularly polarized emission of low energy peaks will no longer be modified by chiral plasmon field.

In summary, we have done additional experiments and demonstrated that for intravalley defect excitons in MoSe₂ monolayer which is unrelated to hybridization, will not exhibit the reversal of DCP. This can prove the hybridization model for intervalley defect excitons is the key to the abnormal reversal of the degree of circular polarization which we have observed in intervalley defect excitons in WSe₂. Therefore, a model considering the hybridization process is crucial and necessary. The hybridization model in our manuscript is effective enough to explain well the experimental results in our manuscript.

Action taken: We have added more descriptions and discussions about the difference between the intervalley defect exciton and intravalley defect exciton in the revised manuscript to show the necessity of the model. (page 6, paragraph 3 with highlighted in red). We also added the additional experimental results of MoSe₂ monolayer in the Supplementary Information. (Supplementary Fig. 18).

2. Moreover, Nature Communications 13, 7691 (2022) has provided experimental evidence for the model that the authors are trying to confirm. As a result, I still maintain that this work is suitable for a more specialized journal.

Reply: We thank the Reviewer for the comments. Although the results of Nature Communications 13, 7691 (2022) provided experimental evidence for hybridization of defect states and dark exciton, **single quantum emitters** have not been investigated. Single quantum emitter is at the heart of quantum optics and photonic quantum-information technologies, so it is crucial to provide direct experimental evidence to prove its intrinsic properties.

We have unambiguously demonstrated the absence of valley polarization and the breaking of valley symmetry in single quantum emitters in WSe₂ by using chiral plasmonic nanocavities, and at the same time demonstrated the validity of the hybridization model (as shown above). Our results provide solid and unambiguous insights on the microscopic origin of quantum emitters in 2D semiconductors and will dramatically change the understanding of these quantum emitters.

Furthermore, we also explained clearly the PL disappearing of the high energy branch of the quantum emitters in magnetic field by the hybridization model, which has not been discussed in detail before.

3. The authors should at least try to compare and contrast the effect of their chiral photonics cavity has on quantum emitters versus free excitons to further support their reasoning.

Reply: We thank the Reviewer for the comments. Due to the fluorescence quenching of free excitons in WSe₂ caused by the nonradiative energy transfer to the Au nanostructure, as we have discussed in the manuscript (page 3, paragraph 1 with highlight in blue), it is hard to make the corresponding comparisons between the effects of chiral plasmonic nanocavity on quantum emitters and free excitons. Instead, we provide more convincing and straightforward

experimental results, that is, a direct comparison between the coupling with intervalley defect excitons and intravalley defect excitons, as shown in Figure R3. This further proves the validity of our conclusion on the absence of valley physics and the validity of intervalley defect exciton model in the quantum emitters in WSe₂ monolayer.

Action taken: We have added more descriptions and discussions about the difference between the intervalley defect exciton and intravalley defect exciton in the revised manuscript to show the validity of the model. (page 6, paragraph 3 with highlighted in red). We also added the additional experimental results of MoSe₂ monolayer in the Supplementary Information. (Supplementary Fig. 18).

REVIEWERS' COMMENTS

Reviewer #3 (Remarks to the Author):

The authors have performed additional measurement which lends some more support to their conclusions. However, I would suggest that the authors dilute the claim that their data provides a smoking gun validation of the theoretical model. It should not appear to a less well-versed reader that the origin of single quantum emitters in TMDs is now completely understood.

Response to Reviewers' Comments and Revisions

Reviewers' comments:

Reviewer #3 (Remarks to the Author):

The authors have performed additional measurement which lends some more support to their conclusions. However, I would suggest that the authors dilute the claim that their data provides a smoking gun validation of the theoretical model. It should not appear to a less well-versed reader that the origin of single quantum emitters in TMDs is now completely understood.

Reply: We thank the Reviewer for the appreciation of our reply and the helpful suggestion. According to the Reviewer's suggestion, we moderate our claims regarding the validity of the hybridization model and the absence of valley symmetry of the emitters.

In the **abstract**, 1) we change the statement "...which **indicates** that the polarization state of emitted photons is modulated by the chiral nanocavity instead of the valley-dependent optical selection rules" to "...**suggests** that the polarization state of emitted photon is modulated by the chiral nanocavity instead of the valley-dependent optical selection rules". 2) We change the statement "Calculations of cavity quantum electrodynamics using the intervalley defect excitons model with spin-valley locking breaking further **confirm** the absence of intrinsic valley polarization" to "Calculations of cavity quantum electrodynamics further **show** the absence of intrinsic valley polarization".

In the **main text**, 1) we change the statement "we experimentally **demonstrate** that the polarization of emitted photons can be dominated by chiral nanocavities, and the DCP (degree of circular polarization) of QEs is independent of transition channels from two different valleys" to "we experimentally **observed** that the polarization states of emitted photons from two different Zeeman splitting peaks are the same for the QEs coupled with chiral plasmonic nanocavities, **suggesting** the degree of circular polarization of the coupled QEs is independent of the transition channels from two different valleys". 2) we change the statement "The absence of valley-dependent optical selection rules in WSe₂ QE can also be **verified** by solving the cavity quantum electrodynamics of the coupled system" to "To investigate the underlying mechanism of the anomalous chiral photon emission from the WSe₂ monolayer, we employed a dynamical solving approach to study the coupled QE-CPR system".

In the **discussion**, we change the statement "we experimentally **demonstrate** that the single quantum emitter in the two-dimensional WSe₂ monolayer breaks the valley symmetry by interacting with the chiral plasmonic nanocavity. The helicities of emitted photons are the same for two different valley transition channels, **indicating** the circularly polarized states of output photons are dominated by the chiral nanocavity instead of the valley-dependent optical selection rules, which also **confirms** the previous theoretical hypothesis about the intervalley defect excitons in WSe₂ monolayer" to "we show experimental **evidence** that the single quantum emitter in the two-dimensional WSe₂ monolayer breaks the valley symmetry by interacting with the chiral plasmonic nanocavity. The helicities of emitted photons remain consistent across two different Zeeman splitting peaks, **suggesting** the circularly polarized

states of output photons are dominated by the chiral nanocavity instead of the valley-dependent optical selection rules. Our results also **show** that the intervalley defect exciton is **one of** the microscopic origins of the QEs in transition-metal dichalcogenides monolayer”.

Action taken: All the changes are highlighted in red in the revised manuscript.